# A Versatile Influence Function for Data Attribution with Non-Decomposable Loss

## Abstract

*Influence function*, a technique rooted in robust statistics, has been adapted in modern machine learning for a novel application: data attribution—quantifying how individual training data points affect a model's predictions. However, the common derivation of influence functions in the data attribution literature is limited to loss functions that can be decomposed into a sum of individual data point losses, with the most prominent examples known as M-estimators. This restricts the application of influence functions to more complex learning objectives, which we refer to as *non-decomposable losses*, such as contrastive or ranking losses, where a unit loss term depends on multiple data points and cannot be decomposed further. In this work, we bridge this gap by revisiting the general formulation of influence function from robust statistics, which extends beyond M-estimators. Based on this formulation, we propose a novel method, the *Versatile Influence Function* (VIF), that can be straightforwardly applied to machine learning models trained with any non-decomposable loss. In comparison to the classical approach in statistics, the proposed VIF is designed to fully leverage the power of auto-differentiation, hereby eliminating the need for case-specific derivations of each loss function. We demonstrate the effectiveness of VIF across three examples: Cox regression for survival analysis, node embedding for network analysis, and listwise learning-to-rank for information retrieval. In all cases, the influence estimated by VIF closely resembles the results obtained by brute-force leave-one-out retraining, while being up to $10^3$ times faster to compute. We believe VIF represents a significant advancement in data attribution, enabling efficient influence-function-based attribution across a wide range of machine learning paradigms, with broad potential for practical use cases.

## 1 Introduction

*Influence function* (IF) is a well-established technique originating from robust statistics and has been adapted to the novel application of *data attribution* in modern machine learning (Koh & Liang, 2017). Data attribution aims to quantify the impact of individual training data points on model outputs, which enables a wide range of data-centric applications such as mislabeled data detection (Koh & Liang, 2017), data selection (Xia et al., 2008), and copyright compensation (Deng & Ma, 2023).

Despite its broad potential, the application of IFs for data attribution has been largely limited to loss functions that can be decomposed into a sum of individual data point losses—such as those commonly used in supervised learning or maximum likelihood estimation, which are also known as M-estimators. This limitation arises from the specific way that IFs are typically derived in the data attribution literature (Koh & Liang, 2017; Grosse et al., 2023), where the derivation involves perturbing the weights of individual data point losses. As a result, this restricts the application of IF-based data attribution methods to more complex machine learning objectives, such as contrastive or ranking losses, where a unit loss term depends on multiple data points and cannot be further decomposed into individual data point losses. We refer to such loss functions as *non-decomposable losses*.

To address this limitation, we revisit the general formulation of IF in statistics literature (Huber & Ronchetti, 2009), which can extend beyond M-estimators. Specifically, statistical estimators are viewed as functionals of probability measures, and the IF is derived as a functional derivative in a

specific perturbation direction. In principle, this formulation applies to any estimator defined as the minimizer of a loss function that depends on an (empirical) probability measure, which corresponds to the learned parameters in the context of machine learning. However, directly applying this general formulation to modern machine learning models poses significant challenges. Firstly, deriving the precise IF for a particular loss function often requires complex, case-by-case mathetical derivations, which can be challenging for intricate loss functions and models. Secondly, for non-convex models, the (local) minimizer of the loss function is not unique; as a result, the mapping from the probability measure to the learned model parameters is not well-defined, making it unclear how the IF should be derived.

To overcome these challenges, we propose the Versatile Influence Function (VIF), a novel method that extends IF-based data attribution to models trained with non-decomposable losses. The proposed VIF serves as an approximation of the general formulation of IF but can be efficiently computed using auto-differentiation tools available in modern machine learning libraries. This approach eliminates the need for case-specific derivations of each loss function. Furthermore, like existing IF-based data attribution methods, VIF does not require model retraining and can be generalized to non-convex models using similar heuristic tricks (Koh & Liang, 2017; Grosse et al., 2023).

We validate the effectiveness of VIF through both theoretical analysis and empirical experiments. In special cases like M-estimation, VIF recovers the classical IF exactly. For Cox regression, we show that VIF closely approximates the classical IF. Empirically, we demonstrate the practicality of VIF across several tasks involving non-decomposable losses: Cox regression for survival analysis, node embedding for network analysis, and listwise learning-to-rank for information retrieval. In all cases, VIF closely approximates the influence obtained from the brute-force leave-one-out retraining while significantly reducing computational time—achieving speed-ups of up to $10^3$ times. We also provide case studies demonstrating VIF can help interpret the behavior of the models. By extending IF to non-decomposable losses, VIF opens new opportunities for data attribution in modern machine learning models, enabling data-centric applications across a wider range of domains.

## 2 RELATED WORK

**Data Attribution.** Data attribution methods can be roughly categorized into two groups: retraining-based and gradient-based methods (Hammoudeh & Lowd, 2024). Retraining-based methods (Ghorbani & Zou, 2019; Jia et al., 2019; Kwon & Zou, 2021; Wang & Jia, 2023; Ilyas et al., 2022) typically estimate the influence of individual training data points by repeatedly retraining models on subsets of the training dataset. While these methods have been shown effective, they are not scalable for large-scale models and applications. In contrast, gradient-based methods (Koh & Liang, 2017; Guo et al., 2020; Barshan et al., 2020; Schioppa et al., 2022; Kwon et al., 2023; Yeh et al., 2018; Pruthi et al., 2020; Park et al., 2023) estimate the training data influence based on the gradient and higher-order gradient information of the original model, avoiding expensive model retraining. In particular, many gradient-based methods (Koh & Liang, 2017; Guo et al., 2020; Barshan et al., 2020; Schioppa et al., 2022; Kwon et al., 2023; Pruthi et al., 2020; Park et al., 2023) can be viewed as variants of IF-based data attribution methods. Therefore, extending IF-based data attribution methods to a wider domains could lead to a significant impact on data attribution.

**Influence Function in Statistics.** The IF is a well-established concept in statistics dating back at least to Hampel (1974), though it is typically applied for purposes other than data attribution. Originally introduced in the context of robust statistics, it was used to assess the robustness of statistical estimators (Huber & Ronchetti, 2009) and later adapted as a tool for developing asymptotic theories (van der Vaart, 2012). Notably, IFs have been derived for a wide range of estimators beyond M-estimators, including L-estimators, R-estimators, and others (Huber & Ronchetti, 2009; van der Vaart, 2012). Closely related to an example of this study, Reid & Crepeau (1985) developed the IF for the Cox regression model. However, the literature in statistics often approaches the derivation of IFs through precise definitions specific to particular estimators, requiring case-specific derivations. In contrast, this work proposes an approximation for the general IF formulation in statistics, which can be straightforwardly applied to a broad family of modern machine learning loss functions for the purpose of data attribution. While this approach involves some degree of approximation, it benefits from being more versatile and computationally efficient, leveraging auto-differentiation capabilities provided by modern machine learning libraries.

## 3 THE VERSATILE INFLUENCE FUNCTION

### 3.1 PRELIMINARIES: IF-BASED DATA ATTRIBUTION FOR DECOMPOSABLE LOSS

We begin by reviewing the formulation of IF-based data attribution in prior literature (Koh & Liang, 2017; Schioppa et al., 2022; Grosse et al., 2023). IF-based data attribution aims to approximate the effect of leave-one-out (LOO) retraining—the change of model parameters after removing one training data point and retraining the model—which could be used to quantify the influence of this training data point.

Formally, suppose we have the following loss function[1],

$$\mathcal{L}_D(\theta) = \sum_{i=1}^n \ell(\theta; z_i), \tag{1}$$

where $\theta$ is the model parameters, $\{z_i\}_{i=1}^n$ is the training dataset, and each $\ell(\cdot; z_i), i = 1, \ldots, n$, corresponds to the loss function of one training data point $z_i$. The IF-based data attribution is derived by first inserting a binary weight $w_i$ in front of each $\ell(\cdot; z_i)$ to represent the inclusion or removal of the individual data points, transforming $\mathcal{L}_D(\theta)$ to a weighted loss

$$\mathcal{L}_D(\theta, w) = \sum_{i=1}^n w_i \ell(\theta; z_i). \tag{2}$$

Note that $w = \mathbf{1}$ corresponds to the original loss in Eq. (1); while removing the $i$-th data point is to set $w_i = 0$ or, equivalently, $w = \mathbf{1}_{-i}$, where $\mathbf{1}_{-i}$ is a vector of all one except for the $i$-th element being zero. Denote the learned parameters as $\hat{\theta}_D(w) := \arg\min_\theta \mathcal{L}_D(\theta, w)$[2]. The LOO effect for data point $i$ is then characterized by $\hat{\theta}_D(\mathbf{1}_{-i}) - \hat{\theta}_D(\mathbf{1})$.

However, evaluating $\hat{\theta}_D(\mathbf{1}_{-i})$ is computationally expensive as it requires model retraining. Koh & Liang (2017) proposed to approximate the LOO effect by relaxing the binary weights in $w$ to the continuous interval $[0, 1]$ and measuring the influence of the training data point $z_i$ on the learned parameters as

$$\left. \frac{\partial \hat{\theta}_D(w)}{\partial w_i} \right|_{w=\mathbf{1}} = -\left[ \nabla_\theta^2 \mathcal{L}_D(\hat{\theta}_D(\mathbf{1}), \mathbf{1}) \right]^{-1} \nabla_\theta \ell(\hat{\theta}_D(\mathbf{1}); z_i), \tag{3}$$

which can be evaluated using only $\hat{\theta}_D(\mathbf{1})$, hence eliminating the need for expensive model retraining.

However, by construction, this approach critically relies on the introduction of the loss weights $w_i$'s, and is thus limited to loss functions that are *decomposable* with respect to the individual training data points, taking the form of Eq. (1).

### 3.2 NON-DECOMPOSABLE LOSS

In practice, there are many common loss functions that are *not* decomposable. Below we list a few examples.

**Example 1: Cox's Partial Likelihood.** The Cox regression model (Cox, 1972) is one of the most widely used models in survival analysis, designed to analyze the time until specific events occur (e.g., patient death or customer churn). A unique challenge in survival analysis is handling *censored* observations, where the exact event time is unknown because the event has either not occurred by the end of the study or the individual is lost to follow-up. These censored data points contain partial information about the event timing and should be properly modeled to improve estimation. The Cox regression model is defined through specifying a hazard function over time $t$ conditional on the individual feature $x$:

$$h(t \mid x) = h_0(t) \exp(\theta^\top x),$$

---

[1]The subscript $D$ in $\mathcal{L}_D$ refers to "decomposable", which is included to differentiate with the later notation.

[2]While this definition is technically valid only under specific assumptions about the loss function (e.g., strict convexity), in practice, methods developed based on these assumptions (together with some heuristics tricks) are often applicable to more complicated models such as neural networks (Koh & Liang, 2017).

where $h_0(t)$ is a baseline hazard function and $\exp(\theta^\top x)$ is the relative risk with $\theta$ as the model parameters to be estimated. Given $n$ data points $\{(X_i, Y_i, \Delta_i)\}_{i=1}^n$, where $X_i$ represents the features for the $i$-th data point, $Y_i$ denotes the observed time (either the event time or the censoring time), and $\Delta_i$ is the binary event indicator ($\Delta_i = 1$ if the event has occurred and $\Delta_i = 0$ if the observation is censored), the parameters $\theta$ can be learned through minimizing the following *negative log partial likelihood*

$$\mathcal{L}_{\text{Cox}}(\theta) = -\sum_{i:\Delta_i=1}\left(\theta^\top X_i - \log\sum_{j\in R_i}\exp(\theta^\top X_j)\right), \tag{4}$$

where $R_i := \{j : Y_j > Y_i\}$ is called the *at-risk set* for the $i$-th data point.

In Eq. (4), each data point may appear in multiple loss terms if it belongs to the at-risk sets of other data points. Consequently, we can no longer characterize the effect of removing a training data point by simply introducing the loss weight.

**Example 2: Contrastive Loss.** Contrastive losses are commonly seen in unsupervised representation learning across various modalities, such as word embeddings (Mikolov et al., 2013), image representations (Chen et al., 2020), or node embeddings (Perozzi et al., 2014). Generally, contrastive losses rely on a set of triplets, $D = \{(u_i, v_i, N_i)\}_{i=1}^m$, where $u_i$ is an anchor data point, $v_i$ is a positive data point that is relevant to $u_i$, while $N_i$ is a set of negative data points that are irrelevant to $u_i$. The contrastive loss is then the summation over such triplets:

$$\mathcal{L}_{\text{Contrast}}(\theta) = \sum_{i=1}^m \ell(\theta; (u_i, v_i, N_i)), \tag{5}$$

where the loss $l(\cdot)$ could take many forms. In word2vec (Mikolov et al., 2013) for word embeddings or DeepWalk (Perozzi et al., 2014) for node embeddings, $\theta$ corresponds to the embedding parameters for each word or node, while the loss $l(\cdot)$ could be defined by heirarchical softmax or negative sampling (see Rong (2014) for more details).

Similar to Eq. (4), each single term of the contrastive loss in Eq. (5) involves multiple data points. Moreover, taking node embeddings as an example, the set of triplets $D$ is constructed by running random walks on the network. Removing one data point, which is a node in this context, could also affect the proximity of other pairs of nodes and hence the construction of $D$.

**Example 3: Listwise Learning-to-Rank.** Learning-to-rank is a core technology underlying information retrieval applications such as search and recommendation. In this context, listwise learning-to-rank methods aim to optimize the ordering of a set of documents or items based on their relevance to a given query. One prominent example of such methods is ListMLE (Xia et al., 2008). Suppose we have annotated results for $m$ queries over $n$ items as a dataset $\{(x_i, (y_i^{(1)}, y_i^{(2)}, \ldots, y_i^{(k)}))\}_{i=1}^m$, where $x_i$ is the query feature, $y_i^{(1)}, y_i^{(2)}, \ldots, y_i^{(k)} \in [n] := \{1, \ldots, n\}$ indicate the top $k$ items for query $i$. Then the ListMLE loss function is defined as following

$$\mathcal{L}_{\text{LTR}}(\theta) = -\sum_{i=1}^m \sum_{j=1}^k \left(f(x_i; \theta)_j - \log \sum_{l \in [n] \setminus \{y_i^{(1)}, \ldots, y_i^{(j-1)}\}} \exp(f(x_i; \theta)_l)\right), \tag{6}$$

where $f(\cdot; \theta)$ is a model parameterized by $\theta$ that takes the query feature as input and outputs $n$ logits for predicting the relevance of the $n$ items.

In this example, Eq. (6) is decomposable with respect to the queries while not decomposable with respect to the items. The influence of items could also be of interest in information retrieval applications. For example, in a search engine, we may want to detect webpages with malicious search engine optimization (Invernizzi et al., 2012); in product co-purchasing recommendation (Zhao et al., 2017), both the queries and items are products.

**A General Loss Formulation.** The examples above can be viewed as special cases of the following formal definition of *non-decomposable loss*.

**Definition 3.1** (Non-Decomposable Loss). *Given $n$ objects of interest within the training data, let a binary vector $b \in \{0, 1\}^n$ indicate the presence of the individual objects in training, i.e., for*

$i = 1, \ldots, n$,

$$b_i = \begin{cases} 1 & \textit{if the } i\textit{-th object presents,} \\ 0 & \textit{otherwise.} \end{cases}$$

*Suppose the machine learning model parameters are denoted as $\theta \in \mathbb{R}^d$, a non-decomposable loss is any function*

$$\mathcal{L} : \mathbb{R}^d \times \{0, 1\}^n \to \mathbb{R},$$

*that maps given model parameters $\theta$ and the object presence vector $b$ to a loss value $\mathcal{L}(\theta, b)$.*

Denoting $\hat{\theta}(b) = \arg\min_\theta \mathcal{L}(\theta, b)$ on any non-decomposable loss $\mathcal{L}(\theta, b)$, the LOO effect of data point $i$ on the learned parameters can still be properly defined by

$$\hat{\theta}(\mathbf{1}_{-i}) - \hat{\theta}(\mathbf{1}).$$

However, in this case, we can no longer use the partial derivative with respect to $b_i$ to approximate the LOO effect, as $\hat{\theta}(b)$ is only well-defined for binary vectors $b$.

**Remark 1** ("Non-Decomposable" v.s. "Not Decomposable"). *The class of non-decomposable loss in Definition 3.1 includes the decomposable loss in Eq. (1) as a special case when $\mathcal{L}(\theta, b) := \sum_{i:b_i=1} l_i(\theta)$. Throughout this paper, we will call loss functions that cannot be written in the form of Eq. (1) as "not decomposable". We name the general class of loss functions in Definition 3.1 as* non-decomposable loss *to highlight that they are generally not decomposable.*

**Remark 2** (Randomness in Losses). *Strictly speaking, many contrastive losses are not deterministic functions of training data points as there is randomness in the construction of the triplet set $D$, due to procedures such as negative sampling or random walk. However, our method derived for the deterministic non-decomposable loss still gives meaningful results in practice for losses with randomness.*

### 3.3 THE STATISTICAL PERSPECTIVE OF INFLUENCE FUNCTION

**The Statistical Formulation of IF.** To derive IF-based data attribution for non-decomposable losses, we revisit a general formulation of IF in robust statistics (Huber & Ronchetti, 2009). Let $\Omega$ be a sample space, and $T(\cdot)$ is a function that maps from a probability measure on $\Omega$ to a vector in $\mathbb{R}^d$. Let $P$ and $Q$ be two probability measures on $\Omega$. The IF of $T(\cdot)$ at $P$ in the direction $Q$ measures the infinitesimal change towards a specific perturbation direction $Q$, which is defined as

$$\mathrm{IF}(T(P); Q) := \lim_{\varepsilon \to 0} \frac{T((1-\varepsilon)P + \varepsilon Q) - T(P)}{\varepsilon}.$$

In the context of machine learning, the learned model parameters, denoted as $\tilde{\theta}(P)$, can be viewed as a function of the data distribution $P$. Specifically, the parameters of the learned model are typically obtained by minimizing a loss function, i.e., $\tilde{\theta}(P) = \arg\min_\theta \tilde{\mathcal{L}}(\theta, P)$. Here, $\tilde{\mathcal{L}}(\theta, P)$ is a loss function that depends on a probability measure $P$, distinguishing it from the non-decomposable loss $\mathcal{L}(\theta, b)$ that depends on the object presence vector $b$.

Assuming the loss is strictly convex and twice-differentiable with respect to the parameters, the learned parameters $\tilde{\theta}(P)$ are then implicitly determined by the following equation

$$\nabla_\theta \tilde{\mathcal{L}}(\tilde{\theta}(P), P) = \mathbf{0}.$$

Moreover, the IF of $\tilde{\theta}(P)$ with a perturbation towards $Q$ is given by[3]

$$\mathrm{IF}(\tilde{\theta}(P); Q) = - \left[ \nabla_\theta^2 \tilde{\mathcal{L}}(\tilde{\theta}(P), P) \right]^{-1} \lim_{\varepsilon \to 0} \frac{\nabla_\theta \tilde{\mathcal{L}}(\tilde{\theta}(P), (1-\varepsilon)P + \varepsilon Q) - \nabla_\theta \tilde{\mathcal{L}}(\tilde{\theta}(P), P)}{\varepsilon}. \quad (7)$$

The advantage of the IF formulation in Eq. (7) is that it can be applied to more general loss functions by properly specifying $P, Q$, and $\tilde{\mathcal{L}}$.

---

[3]See Appendix A.1 for the derivation.

**Example: Application of Eq. (7) to M-Estimators.** As an example, the following Lemma 3.1 states that the IF in Eq. (3) for decomposable loss can be viewed as a special case of the formulation in Eq. (7). This is a well-known result for M-estimators in robust statistics (Huber & Ronchetti, 2009), and the proof of which can be found in Appendix A.2. Intuitively, with the choice of $P, Q$, and $\tilde{\mathcal{L}}$ in Lemma 3.1, $(1 - \varepsilon)P + \varepsilon Q = (1 - \varepsilon)\mathbb{P}_n + \varepsilon \delta_{z_i}$ leads to the effect of upweighting the loss weight of $z_i$ with a small perturbation, which is essentially how the IF in Eq. (3) is derived.

**Lemma 3.1** (IF for M-Estimators). *Eq. (7) reduces to Eq. (3) up to a constant when we specify that 1) $P$ is the empirical distribution $\mathbb{P}_n = \sum_{i=1}^{n} \delta_{z_i}/n$, where $\delta_{z_i}$ is the Dirac measure, i.e., $\mathrm{Pr}(z_i) = 1$ and $\mathrm{Pr}(z_j) = 0, j \neq i$; 2) $Q = \delta_{z_i}$; and 3) $\tilde{\mathcal{L}}(\theta, P) := \mathbb{E}_{z \sim P}[\ell(\theta; z)]$. Specifically,*

$$\mathrm{IF}(\tilde{\theta}(\mathbb{P}_n); \delta_{z_i}) = -n \left[ \nabla_\theta^2 \mathcal{L}_D(\hat{\theta}_D(\mathbf{1}), \mathbf{1}) \right]^{-1} \nabla_\theta \ell(\hat{\theta}_D(\mathbf{1}); z_i).$$

**Challenges of Applying Eq. (7) in Modern Machine Learning.** While the IF in Eq. (7) is a principled and well-established notion in statistics, there are two unique challenges when applying it to modern machine learning models for general non-decomposable losses. Firstly, solving the limit in the right hand side of Eq. (7) requires case-by-case derivation for different loss functions and models, which can be complicated (see an example of IF for the Cox regression (Reid & Crepeau, 1985) in Appendix A.5). Secondly, the mapping $\tilde{\theta}(P)$, hence the limit, are not well-defined for non-convex loss functions as the (local) minimizer is not unique. A similar problem exists in the IF for decomposable loss in Eq. (3) and Koh & Liang (2017) mitigate this problem through heuristic tricks specifically designed for Eq. (3). However, the IF in Eq. (7) is in general more complicated for non-decomposable losses and generalizing it to modern setups like neural networks remains unclear.

### 3.4 VIF as a Finite Difference Approximation

We now derive the proposed VIF method by applying Eq. (7) to the non-decomposable loss while addressing the aforementioned challenges through a *finite-difference approximation*.

**Definition 3.2** (Finite-Difference IF). *Define the finite-difference IF as following*

$$\widehat{\mathrm{IF}}_\varepsilon(\tilde{\theta}(P); Q) := - \left[ \nabla_\theta^2 \tilde{\mathcal{L}}(\tilde{\theta}(P), P) \right]^{-1} \frac{\nabla_\theta \tilde{\mathcal{L}}(\tilde{\theta}(P), (1 - \varepsilon)P + \varepsilon Q) - \nabla_\theta \tilde{\mathcal{L}}(\tilde{\theta}(P), P)}{\varepsilon}, \quad (8)$$

*which approximates the IF in Eq. (7), $\mathrm{IF}(\tilde{\theta}(P); Q)$, by replacing the limit with a finite difference.*

**Observation on M-Estimators.** The proposed VIF method for general non-decomposable losses is motivated by the following observation in the special case for M-estimators.

**Theorem 3.1** (Finite-Difference IF for M-Estimators). *Under the specification of $P = \mathbb{P}_n, Q = \delta_{z_i}$, and $\tilde{\mathcal{L}} = \mathbb{E}_{z \sim P}[\ell(\theta; z)]$ in Lemma 3.1, the IF is identical to the finite-difference IF with $\varepsilon = -\frac{1}{n-1}$, i.e.,*

$$\mathrm{IF}(\tilde{\theta}(\mathbb{P}_n); \delta_{z_i}) = \widehat{\mathrm{IF}}_{-\frac{1}{n-1}}(\tilde{\theta}(\mathbb{P}_n); \delta_{z_i}).$$

*Furthermore, denote $\mathbb{Q}_{n-1}^{(-i)}$ as the empirical distribution where $\mathrm{Pr}(z_i) = 0$ and $\mathrm{Pr}(z_j) = \frac{1}{n-1}, j \neq i$. Then we have*

$$(1 + \frac{1}{n-1})\mathbb{P}_n - \frac{1}{n-1}\delta_{z_i} = \mathbb{Q}_{n-1}^{(-i)}, \quad \widehat{\mathrm{IF}}_{-\frac{1}{n-1}}(\tilde{\theta}(\mathbb{P}_n); \delta_{z_i}) = -(n-1)\widehat{\mathrm{IF}}_1(\tilde{\theta}(\mathbb{P}_n); \mathbb{Q}_{n-1}^{(-i)}).$$

The first part of Theorem 3.1 suggests that, for M-estimators, the limit in $\mathrm{IF}(\tilde{\theta}(\mathbb{P}_n); \delta_{z_i})$ can be exactly replaced by a finite difference with a proper choice of $\varepsilon$. The second part of Theorem 3.1 further shows that we can construct another finite-difference IF, $\widehat{\mathrm{IF}}_1(\tilde{\theta}(\mathbb{P}_n); \mathbb{Q}_{n-1}^{(-i)})$, with a different choice of $Q = \mathbb{Q}_{n-1}^{(-i)}$ and $\varepsilon = 1$, that differs from $\mathrm{IF}(\tilde{\theta}(\mathbb{P}_n); \delta_{z_i})$ only by a constant factor. For the purpose of data attribution, we typically only care about the relative influence among the training data points, so the constant factor does not matter.

**Generalization to General Non-Decomposable Losses.** The benefit of having the form $\widehat{\mathrm{IF}}_1(\tilde{\theta}(\mathbb{P}_n); \mathbb{Q}_{n-1}^{(-i)})$ is that it is straightforward to generalize this formula from M-estimators to general non-decomposable losses. Specifically, noticing that $\mathbb{P}_n$ and $\mathbb{Q}_{n-1}^{(-i)}$ are respectively empirical distribution on the full dataset and the dataset without $z_i$, we can apply this finite-difference IF to any non-decomposable loss through an appropriate definition of $\tilde{\mathcal{L}}$.

**Proposition 3.1** (Finite-Difference IF on Non-Decomposable Loss). *Let $\mathcal{P}(n)$ be the set of uniform distributions supported on subsets of $n$ fixed points $\{z_i\}_{i=1}$. Note that both of the empirical distributions $\mathbb{P}_n$ and $\mathbb{Q}_{n-1}^{(-i)}$ belong to the set $\mathcal{P}(n)$. For any $P \in \mathcal{P}(n)$, denote $b^P \in \{0,1\}^n$ as a binary vector such that $b_i^P = \mathbb{1}[P(z_i) > 0], i = 1, \ldots, n$. Under the following definition of $\tilde{\mathcal{L}}$:*

$$\tilde{\mathcal{L}}(\theta, P) := \mathcal{L}(\theta, b^P),$$

*we have*

$$\widehat{\mathrm{IF}}_1(\tilde{\theta}(\mathbb{P}_n); \mathbb{Q}_{n-1}^{(-i)}) = \left[\nabla_\theta^2 \mathcal{L}(\hat{\theta}(\mathbf{1}), \mathbf{1})\right]^{-1} \nabla_\theta \left(\mathcal{L}(\hat{\theta}(\mathbf{1}), \mathbf{1}) - \mathcal{L}(\hat{\theta}(\mathbf{1}), \mathbf{1}_{-i})\right). \quad (9)$$

**The Proposed VIF.** We propose the following method to approximate the LOO effect for any non-decomposable loss.

**Definition 3.3** (Versatile Influence Function). *The Versatile Influence Function (VIF) that measures the influence of a data object $i$ on the parameters $\hat{\theta}(\mathbf{1})$ learned from a non-decomposable loss $\mathcal{L}$ is defined as following*

$$\mathrm{VIF}(\hat{\theta}(\mathbf{1}); i) := -\left[\frac{1}{n}\nabla_\theta^2 \mathcal{L}(\hat{\theta}(\mathbf{1}), \mathbf{1})\right]^{-1} \nabla_\theta \left(\mathcal{L}(\hat{\theta}(\mathbf{1}), \mathbf{1}) - \mathcal{L}(\hat{\theta}(\mathbf{1}), \mathbf{1}_{-i})\right). \quad (10)$$

The proposed VIF is a variant of Eq. (9), as it can be easily shown that

$$\mathrm{VIF}(\hat{\theta}(\mathbf{1}); i) = -n\widehat{\mathrm{IF}}_1(\tilde{\theta}(\mathbb{P}_n); \mathbb{Q}_{n-1}^{(-i)}).$$

The inclusion of the additional constant factor is motivated by Theorem 3.1 to make it better align with the original IF in Eq. (7). In practice, this definition is also typically more numerically stable as the Hessian is normalized by $\frac{1}{n}$.

**Computational Advantages.** The VIF defined in Eq. (10) enjoys a few computational advantages. Firstly, VIF depends on the parameters only at $\hat{\theta}(\mathbf{1})$ and does not require $\hat{\theta}(\mathbf{1}_{-i})$. Therefore, it does not require model retraining. Secondly, compared to Eq. (7), VIF only involves gradients and the Hessian of the loss, which can be easily obtained through auto-differentiation provided in modern machine learning libraries. Thirdly, VIF can be applied to more complicated models and accelerated with similar heuristic tricks employed by existing IF-based data attribution methods for decomposable losses (Koh & Liang, 2017; Grosse et al., 2023). We have included the results of efficient approximate implementations of VIF based on Conjugate Gradient (CG) and LiSSA (Agarwal et al., 2017; Koh & Liang, 2017) in Appendix C. Finally, note that VIF calculates the difference $\mathcal{L}(\hat{\theta}(\mathbf{1}), \mathbf{1}) - \mathcal{L}(\hat{\theta}(\mathbf{1}), \mathbf{1}_{-i})$ before taking the gradient with respect to the parameters. In some special cases (see, e.g., the decomposable loss case in Section 3.5), taking the difference before the gradient significantly simplifies the computation as the loss terms not involving the $i$-th data object will cancel out.

**Attributing a Target Function.** In practice, we are often interested in attributing certain model outputs or performance. Similar to Koh & Liang (2017), given a target function of interest, $f(z, \theta)$, that depends on both some data $z$ and the model parameter $\theta$, then the influence of a training data point $i$ on this target function can be obtained through the chain rule:

$$\nabla_\theta f(z, \hat{\theta}(\mathbf{1}))^\top \mathrm{VIF}(\hat{\theta}(\mathbf{1}); i). \quad (11)$$

### 3.5 APPROXIMATION QUALITY IN SPECIAL CASES

To provide insights into how accurately the proposed VIF approximates Eq. (7), we examine the following special cases. Although there is no universal guarantee of the approximation quality for all non-decomposable losses, our analysis in these cases suggests that VIF may perform well in many practical applications.

**M-Estimation (Decomposable Loss).** For a decomposable loss, we have $\nabla_\theta \mathcal{L}_D(\hat{\theta}_D(\mathbf{1}), \mathbf{1}) = \sum_{i=1}^n \nabla_\theta \ell(\hat{\theta}_D(\mathbf{1}); z_i)$ and $\nabla_\theta \mathcal{L}_D(\hat{\theta}_D(\mathbf{1}), \mathbf{1}_{-i}) = \sum_{j=1, j\neq i}^n \nabla_\theta \ell(\hat{\theta}_D(\mathbf{1}); z_j)$. In this case, it is straightforward to see that

$$\mathrm{VIF}(\hat{\theta}(\mathbf{1}); i) = -n\left[\nabla_\theta^2 \mathcal{L}_D(\hat{\theta}_D(\mathbf{1}), \mathbf{1})\right]^{-1} \nabla_\theta \ell(\hat{\theta}_D(\mathbf{1}); z_i),$$

which indicates that the VIF here is identical to the IF in Lemma 3.1 without approximation error.

**Cox Regression.** The close-form of the IF for the Cox regression model, obtained by directly solving the limit in Eq. (7) under the Cox regression model, exists in the statistics literature (Reid & Crepeau, 1985), which allows us to characterize the approximation error of the VIF in comparison to the exact solution.

**Theorem 3.2** (Approximation Error under Cox Regression; Informal)**.** *Denote the exact solution by Reid & Crepeau (1985) as* $\mathrm{IF}_{Cox}(\hat{\theta}(\mathbf{1}); i)$ *while the application of VIF on Cox regression as* $\mathrm{VIF}_{Cox}(\hat{\theta}(\mathbf{1}); i)$. *Their difference is bounded as following:*

$$\mathrm{VIF}_{Cox}(\hat{\theta}(\mathbf{1}); i) - \mathrm{IF}_{Cox}(\hat{\theta}(\mathbf{1}); i) = O_p(\frac{1}{n}).$$

Theorem 3.2 suggests that the approximation error of the VIF vanishes when the training data size is large. A formal statement of this result and its proof can be found in Appendix A.5.

## 4 EXPERIMENTS

### 4.1 EXPERIMENTAL SETUP

We conduct experiments on three examples listed in Section 3.2: Cox Regression, Node Embedding, and Listwise Learning-to-Rank. In this section, we present the performance and runtime of VIF compared to brute-force LOO retraining. We also provide two case studies to demonstrate how the influence estimated by VIF can help interpret the behavior of the trained model.

**Datasets and Models.** We evaluate our approach on multiple datasets across different scenarios. For Cox Regression, we use the METABRIC and SUPPORT datasets (Katzman et al., 2018). For both of the datasets, we train a Cox model using the negative log partial likelihood following Eq. (4). For Node Embedding, we use Zachary's Karate network (Zachary, 1977) and train a DeepWalk model (Perozzi et al., 2014). Specifically, we train a two-layer model with one embedding layer and one linear layer optimized via contrastive loss following Eq. (5), where the loss is defined as the negative log softmax. For Listwise Learning-to-Rank, we use the Delicious (Tsoumakas et al., 2008) and Mediamill (Snoek et al., 2006) datasets. We train a linear model using the loss defined in Eq. (6). Please refer to Appendix B for more detailed experiment settings.

**Target Functions.** We apply VIF to estimate the change of a target function, $f(z, \theta)$, before and after a specific data object is excluded from the model training process. Below are our choice of target functions for difference scenarios.

- For Cox Regression, we study how the relative risk function, $f(x_{test}, \theta) = \exp(\theta^\top x_{test})$, of a test object, $x_{test}$, would change if one training object were removed.
- For Node Embedding, we study how the contrastive loss, $f((u, v, N), \theta) = l(\theta; (u, v, N))$, of an arbitrary pair of test nodes, $(u, v)$, would change if a node $w \in N$ were removed from the graph.
- For Listwise Learning-to-Rank, we study how the ListMLE loss of a test query, $f((x_{test}, y_{test}^{[k]}), \theta) = -\sum_{j=1}^{k} \left( f(x_{test}; \theta)_j - \log \sum_{l \in [n] \setminus \{y_{test}^{(1)}, ..., y_{test}^{(j-1)}\}} \exp(f(x_{test}; \theta)_l) \right)$, would change if one item $l \in [n]$ were removed from the training process.

### 4.2 PERFORMANCE

We utilize the Pearson correlation coefficient to quantitatively evaluate how closely the influence estimated by VIF aligns with the results obtained by brute-force LOO retraining. Furthermore, as a reference upper limit of performance, we evaluate the correlation between two brute-force LOO retraining with different random seeds. As noted in Remark 2, some examples like contrastive losses are not deterministic, which could impact the observed correlations.

Table 1 presents the Pearson correlation coefficients comparing VIF with brute-force LOO retraining using different random seeds. The performance of VIF matches the brute-force LOO in all experimental settings. Except for the Node Embedding scenario, the Pearson correlation coefficients are close to 1, indicating a strong resemblance between the VIF estimates and the retraining results. In the Node Embedding scenario, the correlations are moderately high for both methods due to the

Table 1: The Pearson correlation coefficients of VIF and brute-force LOO retraining under different experimental settings. Specifically, "Brute-Force" refers to the results of two times of brute-force LOO retraining using different random seeds, which serves as a reference upper limit of performance.

| Scenario | Dataset | Method | Pearson Correlation |
|---|---|---|---|
| Cox Regression | METABRIC | VIF | 0.997 |
| | | Brute-Force | 0.997 |
| | SUPPORT | VIF | 0.943 |
| | | Brute-Force | 0.955 |
| Node Embedding | Karate | VIF | 0.407 |
| | | Brute-Force | 0.419 |
| Listwise Learning-to-Rank | Mediamill | VIF | 0.823 |
| | | Brute-Force | 0.999 |
| | Delicious | VIF | 0.906 |
| | | Brute-Force | 0.999 |

inherent randomness in the random walk procedure for constructing the triplet set in the DeepWalk algorithm. Nevertheless, VIF achieves a correlation that is close to the upper limit by brute-force LOO retraining.

### 4.3 RUNTIME

We report the runtime of VIF and brute-force LOO retraining in Tabel 2. The computational advantage of VIF is significant, reducing the runtime by factors up to $1097\times$. This advantage becomes more pronounced as the dataset size increases. The improvement ratio on the Karate dataset is moderate due to the overhead from the random walk process and potential optimizations in the implementation. All runtime measurements were recorded using an Intel(R) Xeon(R) Gold 6338 CPU.

Table 2: Runtime comparison of VIF and brute-force LOO retraining.

| Senario | Dataset | Brute-Force | VIF | Improvement Ratio |
|---|---|---|---|---|
| Cox Regression | METABRIC | 24 min | 2.43 sec | $593\times$ |
| | SUPPORT | 225 min | 12.3 sec | $1097\times$ |
| Network Embedding | Karate | 204 min | 109 min | $1.87\times$ |
| Listwise Learning-to-Rank | Mediamill | 52 min | 2.6 min | $20\times$ |
| | Delicious | 660 min | 2.8 min | $236\times$ |

### 4.4 CASE STUDIES

We present two case studies to show how the influence estimated by VIF can help interpret the behavior of the trained model.

**Case study 1: Cox Regression.** In Table 3, we show the top-5 most influential training samples, as estimated by VIF, for the relative risk function of two randomly selected test samples. We observe that removing two types of data samples in training will significantly increase the relative risk function of a test sample: (1) training samples that share similar features with the test sample and have long survival times (e.g., training sample ranks 1, 3, 4, 5 for test sample 0 and ranks 5 for test sample 1) and (2) training samples that differ in features from the test sample and have short survival times (e.g., training sample ranks 2 for test sample 0 and ranks 1, 2, 3, 4 for test sample 1). These findings align with domain knowledge.

**Case study 2: Node Embedding.** in Figure 1b and 1c, we show the influence of all nodes to the contrastive loss of 2 pairs of test nodes. The spring layout of the Karate dataset is provided in Figure 1a. We observe that the most influential nodes (on the top right in Figure 1b and 1c) are the hub nodes that lie on the shortest path of the pair of test nodes. For example, the shortest path

Table 3: The top-5 influential training samples to 2 test samples in the METABRIC dataset. "Features Similarity" is the cosine similarity between the feature of the influential training sample and the test sample. "Observed Time" and "Event Occurred" are the $Y$ and $\Delta$ of the influential training sample as defined in Eq. (4).

| Influence Rank | Test Sample 0 | | | Test Sample 1 | | |
|---|---|---|---|---|---|---|
| | Feature Similarity | Observed Time | Event Occurred | Feature similarity | Observed time | Event occurred |
| 1 | 0.84 | 322.83 | False | -0.49 | 16.57 | True |
| 2 | -0.34 | 9.13 | True | -0.22 | 30.97 | True |
| 3 | 0.77 | 258.17 | True | -0.39 | 15.07 | True |
| 4 | 0.23 | 131.27 | False | -0.65 | 4.43 | True |
| 5 | 0.81 | 183.43 | False | 0.72 | 307.63 | False |

from node 12 to node 10 passes through node 0, while the shortest path from node 15 to node 13 passes through node 33. Conversely, the nodes with the most negative influence (on the bottom left in Figure 1b and 1c) are those that likely "distract" the random walk away from the test node pairs. For instance, node 3 distracts the walk from node 12 to node 10, and node 30 distracts the walk from node 15 to node 13.

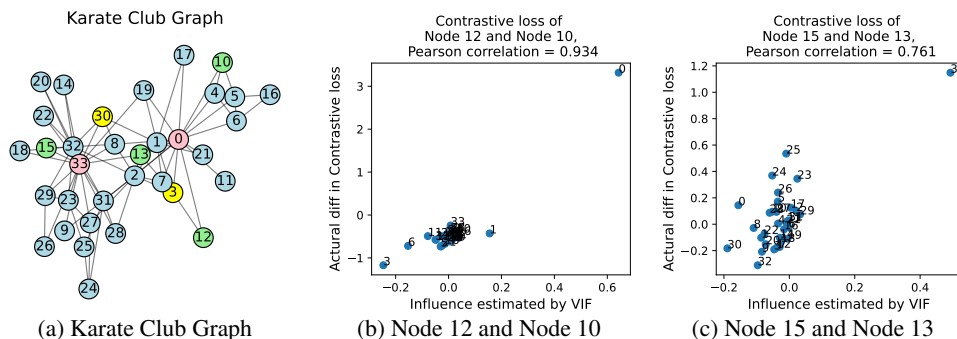

| (a) Karate Club Graph | (b) Node 12 and Node 10 | (c) Node 15 and Node 13 |

Figure 1: VIF is applied to Zachary's Karate network to estimate the influence of each node on the contrastive loss of a pair of test nodes. Figure 1a is a spring layout of the Karate network. Figure 1b and Figure 1c illustrate the alignment between the influence estimated by VIF (x-axis) and the brute-force LOO retrained loss difference (y-axis).

## 5 CONCLUSION

In this work, we introduced the Versatile Influence Function (VIF), a novel method that extends IF-based data attribution to models trained with non-decomposable losses. The key idea behind VIF is a finite difference approximation of the general IF formulation in the statistics literature, which eliminates the need for case-specific derivations and can be efficiently computed with the auto-differentiation tools provided in modern machine learning libraries. Our theoretical analysis demonstrates that VIF accurately recovers classical influence functions in the case of M-estimators and provides strong approximations for more complex settings such as Cox regression. Empirical evaluations across various tasks show that VIF closely approximates the influence obtained by brute-force leave-one-out retraining while being orders-of-magnitude faster. By broadening the scope of IF-based data attribution to non-decomposable losses, VIF opens new avenues for data-centric applications in machine learning, empowering practitioners to explore data attribution in more complex and diverse domains.

**Limitation and Future Work.** Similar to early IF-based methods for decomposable loss (Koh & Liang, 2017), the formal derivation of VIF assumes convexity of the loss function, which requires practical tricks to adapt the proposed method to large-scale neural network models. While we have explored the application of Conjugate Gradient and LiSSA (Agarwal et al., 2017) for efficient inverse Hessian approximation (see Appendix C), more advanced techniques to stabilize and accelerate IF-based methods developed for decomposable losses, such as EK-FAC (Grosse et al., 2023), ensemble (Park et al., 2023), or gradient projection (Choe et al., 2024), may be adapted to further enhance the practical applicability of VIF on large-scale models.

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

## A  OMITTED DERIVATIONS

### A.1  DERIVATION OF EQ. (7)

Consider an $\varepsilon$ perturbation towards another distribution $Q$, i.e., $(1 - \varepsilon)P + \varepsilon Q$. Note that $\tilde{\theta}((1 - \varepsilon)P + \varepsilon Q)$ solves $\nabla_\theta \tilde{\mathcal{L}}(\theta, (1 - \varepsilon)P + \varepsilon Q) = 0$. We take derivative with respect to $\varepsilon$ and evaluate at $\varepsilon = 0$ on both side, which leads to

$$\nabla_\theta^2 \tilde{\mathcal{L}}(\tilde{\theta}(P), P) \lim_{\varepsilon \to 0} \frac{\tilde{\theta}((1 - \varepsilon)P + \varepsilon Q) - \tilde{\theta}(P)}{\varepsilon} + \lim_{\varepsilon \to 0} \frac{\nabla_\theta \tilde{\mathcal{L}}(\tilde{\theta}(P), (1 - \varepsilon)P + \varepsilon Q) - \nabla_\theta \tilde{\mathcal{L}}(\tilde{\theta}(P), P)}{\varepsilon} = 0.$$

Given the strict convexity, the Hessian is invertible at the global optimal. By plugging the definition of IF, we have

$$IF(\tilde{\theta}(P); Q) = -\left[\nabla_\theta^2 \tilde{\mathcal{L}}(\tilde{\theta}(P), P)\right]^{-1} \lim_{\varepsilon \to 0} \frac{\nabla_\theta \tilde{\mathcal{L}}(\tilde{\theta}(P), (1 - \varepsilon)P + \varepsilon Q) - \nabla_\theta \tilde{\mathcal{L}}(\tilde{\theta}(P), P)}{\varepsilon}.$$

### A.2  PROOF OF LEMMA 3.1

*Proof.* Under M-estimation, the objective function becomes the empirical loss, i.e., $\tilde{\mathcal{L}}(\theta, P) = \mathbb{E}_{z \sim P}[\ell(\theta; z)]$, where $P = \mathbb{P}_n = \sum_{i=1}^n \delta_{z_i}/n$ is the empirical distribution over the dataset. Note that $\tilde{\mathcal{L}}(\theta, P) = \frac{1}{n} \mathcal{L}_D(\theta, \mathbf{1})$ for any $\theta$, therefore they share the same minimizer, i.e.,

$$\tilde{\theta}(P) = \hat{\theta}_D(\mathbf{1}).$$

The gradient and Hessian of $\tilde{\mathcal{L}}(\tilde{\theta}(P), P)$ are respectively

$$\nabla_\theta \tilde{\mathcal{L}}(\tilde{\theta}(P), P) = \mathbb{E}_{z \sim P}[\nabla_\theta \ell(\tilde{\theta}(P); z)] = \frac{1}{n} \sum_{j=1}^n \nabla_\theta \ell(\tilde{\theta}(P); z_j) = 0$$

and

$$\nabla_\theta^2 \tilde{\mathcal{L}}(\tilde{\theta}(P), P) = \mathbb{E}_{z \sim P}[\nabla_\theta^2 \ell(\tilde{\theta}(P); z)] = \sum_{i=1}^n \nabla_\theta^2 \ell(\tilde{\theta}(P); z_i)/n = \frac{1}{n} \nabla_\theta^2 \mathcal{L}_D(\hat{\theta}_D(\mathbf{1}), \mathbf{1}).$$

The infinitesimal change on the gradient towards the distribution $Q = \delta_{z_i}$ equals to

$$\lim_{\varepsilon \to 0} \frac{\nabla_\theta \tilde{\mathcal{L}}(\tilde{\theta}(P), (1 - \varepsilon)P + \varepsilon Q) - \nabla_\theta \tilde{\mathcal{L}}(\tilde{\theta}(P), P)}{\varepsilon}$$
$$= \lim_{\varepsilon \to 0} \frac{\mathbb{E}_{z \sim (1 - \varepsilon)P + \varepsilon Q}[\nabla_\theta \ell(\tilde{\theta}(P), z)] - 0}{\varepsilon}$$
$$= \lim_{\varepsilon \to 0} \frac{(1 - \varepsilon)\mathbb{E}_{z \sim P}[\nabla_\theta \ell(\tilde{\theta}(P), z)] + \varepsilon \mathbb{E}_{z \sim Q}[\nabla_\theta \ell(\tilde{\theta}(P), z)]}{\varepsilon}$$
$$= \lim_{\varepsilon \to 0} \frac{(1 - \varepsilon) \cdot 0 + \varepsilon \mathbb{E}_{z \sim Q}[\nabla_\theta \ell(\tilde{\theta}(P), z)]}{\varepsilon}$$
$$= \mathbb{E}_{z \sim Q}[\nabla_\theta \ell(\tilde{\theta}(P), z)]$$
$$= \nabla_\theta \ell(\tilde{\theta}(P), z_i) = \nabla_\theta \ell(\hat{\theta}_D(\mathbf{1}), z_i).$$

Plugging the above equations into Eq. (7), it becomes

$$\text{IF}(\tilde{\theta}(\mathbb{P}_n); \delta_{z_i}) = -n \left[\nabla_\theta^2 \mathcal{L}_D(\hat{\theta}_D(\mathbf{1}), \mathbf{1})\right]^{-1} \nabla_\theta \ell(\hat{\theta}_D(\mathbf{1}); z_i).$$

$\square$

## A.3 PROOF OF THEOREM 3.1

**Lemma A.1.** *Let $\mathbb{P}_n$ and $\mathbb{Q}_{n-1}^{(-i)}$ be the empirical distributions respectively on $\{z_j\}_{j=1}^n$ and $\{z_j\}_{j=1}^n \setminus \{z_i\}$, while $\delta_{z_i}$ is the distribution concentrated on $z_i$. Then*

$$(1 + \frac{1}{n-1})\mathbb{P}_n - \frac{1}{n-1}\delta_{z_i} = \mathbb{Q}_{n-1}^{(-i)}.$$

*Proof of Lemma A.1.* For any $j \neq i$,

$$(1 + \frac{1}{n-1})\mathbb{P}_n(z_j) - \frac{1}{n-1}\delta_{z_i}(z_j) = (1 + \frac{1}{n-1}) \cdot \frac{1}{n} - 0$$
$$= \frac{1}{n-1}$$
$$= \mathbb{Q}_{n-1}^{(-i)}(z_j).$$

For $i$,

$$(1 + \frac{1}{n-1})\mathbb{P}_n(z_i) - \frac{1}{n-1}\delta_{z_i}(z_i) = (1 + \frac{1}{n-1}) \cdot \frac{1}{n} - \frac{1}{n-1} \cdot 1$$
$$= 0$$
$$= \mathbb{Q}_{n-1}^{(-i)}(z_i).$$

$\square$

*Proof of Theorem 3.1.* We first prove the first part of Theorem 3.1, where our goal is to show

$$\widehat{\mathrm{IF}}_{-\frac{1}{n-1}}(\tilde{\theta}(\mathbb{P}_n); \delta_{z_i}) = -n\left[\nabla_\theta^2 \mathcal{L}_D(\hat{\theta}_D(\mathbf{1}), \mathbf{1})\right]^{-1} \nabla_\theta \ell(\hat{\theta}_D(\mathbf{1}); z_i).$$

Expanding $\widehat{\mathrm{IF}}_\varepsilon(\tilde{\theta}(\mathbb{P}_n); \delta_{z_i})$ by its definition in Eq. (8),

$$\widehat{\mathrm{IF}}_\varepsilon(\tilde{\theta}(\mathbb{P}_n); \delta_{z_i}) = -\left[\nabla_\theta^2 \tilde{\mathcal{L}}(\tilde{\theta}(\mathbb{P}_n), \mathbb{P}_n)\right]^{-1} \frac{\nabla_\theta \tilde{\mathcal{L}}(\tilde{\theta}(\mathbb{P}_n), (1-\varepsilon)\mathbb{P}_n + \varepsilon\delta_{z_i}) - \nabla_\theta \tilde{\mathcal{L}}(\tilde{\theta}(\mathbb{P}_n), \mathbb{P}_n)}{\varepsilon}.$$

Setting $\varepsilon = -\frac{1}{n-1}$ and by Lemma A.1,

$$\widehat{\mathrm{IF}}_{-\frac{1}{n-1}}(\tilde{\theta}(\mathbb{P}_n); \delta_{z_i}) = -\left[\nabla_\theta^2 \tilde{\mathcal{L}}(\tilde{\theta}(\mathbb{P}_n), \mathbb{P}_n)\right]^{-1} \frac{\nabla_\theta \tilde{\mathcal{L}}(\tilde{\theta}(\mathbb{P}_n), \mathbb{Q}_{n-1}^{(-i)}) - \nabla_\theta \tilde{\mathcal{L}}(\tilde{\theta}(\mathbb{P}_n), \mathbb{P}_n)}{-1/(n-1)} \quad (12)$$

$$= -\left[\nabla_\theta^2 \tilde{\mathcal{L}}(\tilde{\theta}(\mathbb{P}_n), \mathbb{P}_n)\right]^{-1} \frac{\mathbb{E}_{z \sim \mathbb{Q}_{n-1}^{(-i)}}[\nabla_\theta \ell(\tilde{\theta}(\mathbb{P}_n); z)] - \mathbb{E}_{z \sim \mathbb{P}_n}[\nabla_\theta \ell(\tilde{\theta}(\mathbb{P}_n); z)]}{-1/(n-1)}$$

$$= -\left[\nabla_\theta^2 \tilde{\mathcal{L}}(\tilde{\theta}(\mathbb{P}_n), \mathbb{P}_n)\right]^{-1} \frac{\sum_{j=1, j\neq i}^n \nabla_\theta \ell(\tilde{\theta}(\mathbb{P}_n); z_j)/(n-1) - \sum_{j=1}^n \nabla_\theta \ell(\tilde{\theta}(\mathbb{P}_n); z_j)/n}{-1/(n-1)}$$

$$= -\left[\nabla_\theta^2 \tilde{\mathcal{L}}(\tilde{\theta}(\mathbb{P}_n), \mathbb{P}_n)\right]^{-1} \left[-\sum_{j=1, j\neq i}^n \nabla_\theta \ell(\tilde{\theta}(\mathbb{P}_n); z_j) + \frac{n-1}{n}\sum_{j=1}^n \nabla_\theta \ell(\tilde{\theta}(\mathbb{P}_n); z_j)\right].$$
$$(13)$$

Noting that $\tilde{\theta}(\mathbb{P}_n)$ is the optimizer for $\tilde{\mathcal{L}}(\theta, \mathbb{P}_n)$, so

$$0 = \nabla_\theta \tilde{\mathcal{L}}(\tilde{\theta}(\mathbb{P}_n), \mathbb{P}_n) = \frac{1}{n}\sum_{j=1}^n \nabla_\theta \ell(\tilde{\theta}(\mathbb{P}_n); z_j).$$

Therefore,

$$-\sum_{j=1, j\neq i}^n \nabla_\theta \ell(\tilde{\theta}(\mathbb{P}_n); z_j) = \nabla_\theta \ell(\tilde{\theta}(\mathbb{P}_n); z_i).$$

Plugging the two equations above into Eq. (13), we have

$$\widehat{\mathrm{IF}}_{-\frac{1}{n-1}}(\tilde{\theta}(\mathbb{P}_n); \delta_{z_i}) = - \left[\nabla_\theta^2 \tilde{\mathcal{L}}(\tilde{\theta}(\mathbb{P}_n), \mathbb{P}_n)\right]^{-1} \nabla_\theta \ell(\tilde{\theta}(\mathbb{P}_n); z_i).$$

From the proof of Lemma 3.1 in Appendix A.2, we know

$$\tilde{\theta}(\mathbb{P}_n) = \hat{\theta}_D(\mathbf{1}), \quad \nabla_\theta^2 \tilde{\mathcal{L}}(\tilde{\theta}(\mathbb{P}_n), \mathbb{P}_n) = \frac{1}{n}\nabla_\theta^2 \mathcal{L}_D(\hat{\theta}_D(\mathbf{1}), \mathbf{1}).$$

Therefore,

$$\widehat{\mathrm{IF}}_{-\frac{1}{n-1}}(\tilde{\theta}(\mathbb{P}_n); \delta_{z_i}) = -n \left[\nabla_\theta^2 \mathcal{L}_D(\hat{\theta}_D(\mathbf{1}), \mathbf{1})\right]^{-1} \nabla_\theta \ell(\hat{\theta}_D(\mathbf{1}); z_i),$$

which completes the proof for the first part of Theorem 3.1. For the second part, the first equation has been proved as Lemma A.1. The second equation is straightforward from Eq. (12):

$$\widehat{\mathrm{IF}}_{-\frac{1}{n-1}}(\tilde{\theta}(\mathbb{P}_n); \delta_{z_i}) = - \left[\nabla_\theta^2 \tilde{\mathcal{L}}(\tilde{\theta}(\mathbb{P}_n), \mathbb{P}_n)\right]^{-1} \frac{\nabla_\theta \tilde{\mathcal{L}}(\tilde{\theta}(\mathbb{P}_n), \mathbb{Q}_{n-1}^{(-i)}) - \nabla_\theta \tilde{\mathcal{L}}(\tilde{\theta}(\mathbb{P}_n), \mathbb{P}_n)}{-1/(n-1)}$$

$$= -(n-1)\left( - \left[\nabla_\theta^2 \tilde{\mathcal{L}}(\tilde{\theta}(\mathbb{P}_n), \mathbb{P}_n)\right]^{-1} \frac{\nabla_\theta \tilde{\mathcal{L}}(\tilde{\theta}(\mathbb{P}_n), \mathbb{Q}_{n-1}^{(-i)}) - \nabla_\theta \tilde{\mathcal{L}}(\tilde{\theta}(\mathbb{P}_n), \mathbb{P}_n)}{1} \right)$$

$$= -(n-1)\widehat{\mathrm{IF}}_1(\tilde{\theta}(\mathbb{P}_n); \mathbb{Q}_{n-1}^{(-i)}).$$

$\square$

## A.4 PROOF OF PROPOSITION 3.1

*Proof.* It is easy to verify that

$$b^{\mathbb{P}_n} = \mathbf{1}, \quad b^{\mathbb{Q}_{n-1}^{(-i)}} = \mathbf{1}_{-i}.$$

Hence, based on the definition of $\tilde{\mathcal{L}}$ in Proposition 3.1, we have

$$\tilde{\mathcal{L}}(\theta, \mathbb{P}_n) = \mathcal{L}(\theta, \mathbf{1}), \quad \tilde{\mathcal{L}}(\theta, \mathbb{Q}_{n-1}^{(-i)}) = \mathcal{L}(\theta, \mathbf{1}_{-i}).$$

Therefore, we also have $\tilde{\theta}(\mathbb{P}_n) = \hat{\theta}(\mathbf{1})$. The result in Eq. (9) follows directly by plugging these quantities into the definition of $\widehat{\mathrm{IF}}_1(\tilde{\theta}(\mathbb{P}_n); \mathbb{Q}_{n-1}^{(-i)})$. $\square$

## A.5 FORMAL STATEMENT AND PROOF OF THEOREM 3.2

**Setup and Notation.** For convenience, we adopt a set of slightly different notations tailored for the Cox regression model. Consider $n$ i.i.d. generated right-censoring data $\{Z_i = (X_i, Y_i, \Delta_i)\}_{i=1}^n$, where $Y_i = min\{T_i, C_i\}$ is the observed time, $T_i$ is the time to event of interest, and $C_i$ is the censoring time. We assume non-informative censoring, i.e., $T$ and $C_i$ are independent conditional on $X$, which is a common assumption in the literature. Suppose there are no tied events for simplicity.

A well-known estimate for the coefficients $\beta$ under the Cox model is obtained by minimizing the negative log-partial likelihood:

$$\mathcal{L}_n(\theta) := -\sum_{i=1}^n \Delta_i \left( \theta^\top X_i - \log\left( \sum_{j \in R_i} \exp\left(\theta^\top X_j\right) \right) \right)$$

$$= -\sum_{i=1}^n \Delta_i \left( \theta^\top X_i - \log\left( \sum_{j=1}^n I(Y_j \geq Y_i) \exp\left(\theta^\top X_j\right) \right) \right).$$

Note that $\mathcal{L}_n(\theta)$ is a convex function and the estimate $\hat{\theta}$ equivalently solves the following *score equation*:

$$\nabla_\theta \mathcal{L}_n(\hat{\theta}) = \sum_{i=1}^n \underbrace{-\Delta_i \left( X_i - \frac{S_n^{(1)}(Y_i; \hat{\theta})}{S_n^{(0)}(Y_i; \hat{\theta})} \right)}_{\nabla_\theta \ell_n(\hat{\theta}; Z_i)} = 0,$$

where

$$S_n^{(0)}(t;\theta) = \frac{1}{n}\sum_{i=1}^n I\left(Y_i \geq t\right)\exp\left(\theta^\top X_i\right), \tag{14}$$

$$S_n^{(1)}(t;\theta) = \frac{1}{n}\sum_{i=1}^n I\left(Y_i \geq t\right)\exp\left(\theta^\top X_i\right)X_i. \tag{15}$$

It has been shown that, under some regularity conditions, $\hat\theta$ is a consistent estimator for $\theta^*$. Note that the above score equation is not a simple estimation equation that takes the summation of i.i.d. terms, because $S_n^{(0)}(t;\theta)$ and $S_n^{(1)}(t;\theta)$ depend on all observations.

**Analytical Form of Influence Function in Statistics.** Reid & Crepeau (1985) derived the influence function by evaluating the limit in (7) with $P$ being the underlying data-generating distribution and $Q = \delta_{Z_i}$ (i.e., the Gateaux derivative at $\theta^* = \theta(P)$ in the direction $\delta_{Z_i}$). To start with, we define the population counterparts of Eq. (14) and Eq. (15):

$$s^{(0)}(t;\theta) = \mathbb{E}\left(I\left(Y \geq t\right)\exp\left(\theta^\top X\right)\right),$$
$$s^{(1)}(t;\theta) = \mathbb{E}\left(I\left(Y \geq t\right)\exp\left(\theta^\top X\right)X\right),$$

and introduce the counting process notation: the counting process associated with $i$-th data $N_i(t) = I(Y_i \leq t, \Delta_i = 1)$, the process $G_n(t) = \frac{1}{n}\sum_{i=1}^n N_i(t)$, and its population expectation $G(t) = \mathbb{E}(G_n(t))$. Then the influence function for the observation $Z_i = (X_i, Y_i, \Delta_i)$ is given by

$$\boldsymbol{A}\cdot\mathrm{IF}(i) = \Delta_i\left(X_i - \frac{s^{(1)}(Y_i;\theta^*)}{s^{(0)}(Y_i;\theta^*)}\right)$$
$$- \exp(\theta^{*\top}X_i)\cdot\int\frac{I(Y_i \geq t)}{s^{(0)}(t;\theta^*)}\left(X_i - \frac{s^{(1)}(t;\theta^*)}{s^{(0)}(t;\theta^*)}\right)dG(t)$$

where $\boldsymbol{A}$ is the non-singular information matrix. A consistent estimate for $\boldsymbol{A}$ is given by $\nabla_\theta^2\mathcal{L}_n(\hat\theta)/n$. The empirical influence function given $n$ data points is obtained by substituting $\boldsymbol{A}$, $\theta^*$, and $G(t)$ by $\nabla_\theta^2\mathcal{L}(\hat\theta)/n$, $\hat\theta$, and $G_n(t)$ respectively:

$$\mathrm{IF}_n(i) = -\left[\nabla_\theta^2\mathcal{L}(\hat\theta)/n\right]^{-1}\nabla_\theta\ell_n(\hat\theta;Z_i) - \left[\nabla_\theta^2\mathcal{L}(\hat\theta)/n\right]^{-1}C_i(\hat\theta),$$

where

$$C_i(\hat\theta) = \exp(\hat\theta^\top X_i)\cdot\frac{1}{n}\sum_{j=1}^n\int\frac{I(Y_i \geq t)}{S_n^{(0)}(t;\hat\theta)}\left(X_i - \frac{S_n^{(1)}(t;\hat\theta)}{S_n^{(0)}(t;\hat\theta)}\right)dN_j(t)$$
$$= \exp(\hat\theta^\top X_i)\cdot\frac{1}{n}\sum_{j=1}^n\frac{I(Y_i \geq Y_j)\Delta_j}{S_n^{(0)}(Y_j;\hat\theta)}\cdot\left(X_i - \frac{S_n^{(1)}(Y_j;\hat\theta)}{S_n^{(0)}(Y_j;\hat\theta)}\right).$$

The first term is analogous to the standard influence function for M-estimators and the second term captures the influence of the $i$-th observation in the at-risk set.

**The Proposed VIF.** Under the Cox regression, the proposed VIF becomes

$$\mathrm{VIF}_n(i) := -\left[\nabla_\theta^2\mathcal{L}_n(\hat\theta)/n\right]^{-1}\left(\nabla_\theta\mathcal{L}_n(\hat\theta) - \nabla_\theta\mathcal{L}_{n-1}^{(-i)}(\hat\theta)\right),$$

where $\nabla_\theta\mathcal{L}_{n-1}^{(-i)}(\hat\theta)$ is the gradient of the negative log-partial likelihood after excluding the $i$-th data point at $\hat\theta$. Given no tied events, we can rewrite $\nabla_\theta\mathcal{L}_{n-1}^{(-i)}(\hat\theta)$ as

$$\nabla_\theta\mathcal{L}_{n-1}^{(-i)}(\hat\theta) = -\sum_{j:Y_j<Y_i}\Delta_j\left(X_j - \frac{S_n^{(1)}(Y_j;\hat\theta) - \exp(\hat\theta^\top X_i)X_i/n}{S_n^{(0)}(Y_j;\hat\theta) - \exp(\hat\theta^\top X_i)/n}\right) - \sum_{j:Y_j>Y_i}\Delta_j\left(X_j - \frac{S_n^{(1)}(Y_j;\hat\theta)}{S_n^{(0)}(Y_j;\hat\theta)}\right).$$

Then it follows that

$$\mathrm{VIF}_n(i) = -[\nabla_\theta^2\mathcal{L}_n(\hat\theta)/n]^{-1}\left(\nabla_\theta\mathcal{L}_n(\hat\theta) - \nabla_\theta\mathcal{L}_{n-1}^{(-i)}(\hat\theta)\right)$$

$$
\begin{aligned}
= & -[\nabla_\theta^2 \mathcal{L}_n(\hat\theta)/n]^{-1} \left( \nabla_\theta \ell_n(\hat\theta; Z_i) + \sum_{j:Y_j < Y_i} \Delta_j \left( \frac{S_n^{(1)}(Y_j; \hat\theta)}{S_n^{(0)}(Y_j; \hat\theta)} - \frac{S_n^{(1)}(Y_j; \hat\theta) - \exp(\hat\theta^\top X_i)X_i/n}{S_n^{(0)}(Y_j; \hat\theta) - \exp(\hat\theta^\top X_i)/n} \right) \right) \\
= & -[\nabla_\theta^2 \mathcal{L}_n(\hat\theta)/n]^{-1} \nabla_\theta \ell_n(\hat\theta; Z_i) \\
& -[\nabla_\theta^2 \mathcal{L}_n(\hat\theta)/n]^{-1} \left( \exp(\hat\theta^\top X_i) \cdot \frac{1}{n} \sum_{j=1}^n \frac{I(Y_j < Y_i)\Delta_j}{S_n^{(0)}(Y_j; \hat\theta) - \exp(\hat\theta^\top X_i)/n} \cdot \left( X_i - \frac{S_n^{(1)}(Y_j; \hat\theta)}{S_n^{(0)}(Y_j; \hat\theta)} \right) \right).
\end{aligned}
$$

**Approximation Error.** Below, we formally bound the difference between the analytical form of IF and our proposed approximation. Our result implies that the difference between the analytic expression of the IF and the proposed VIF approximation, i.e., $\mathrm{VIF}_n(i) - \mathrm{IF}_n(i)$, diminishes at a rate of $1/n$ as the sample size grows and is of a smaller order than $\mathrm{IF}_n(i)$. This is because $\mathrm{IF}_n(i) = \mathrm{IF}(i) + o_p(1)$, where $\mathrm{IF}(i)$ is a non-degenerate random variable that doesn't converge to zero in probability; therefore $\mathrm{IF}_n(i)$ remains bounded away from zero in probability, denoted as $= \Omega_p(1)$.

**Theorem A.1** (Approximation Error Bound under Cox Model). *Assume that (1) the true parameter $\theta^*$ is an interior point of a compact set $\mathcal{B} \subset \mathbf{R}^d$; (2) the density of $X$ is bounded below by a constant $c > 0$ over its domain $\mathcal{X}$, which is a compact subset of $\mathbf{R}^d$; (3) there is a truncation time $\tau < \infty$ such that for some constant $\delta_0$, $\Pr(Y > \tau | X) \geq \delta_0$ almost surely; (4) the information matrix $\mathbf{A}$ is non-singular. Assuming uninformative censoring, the difference between $\mathrm{IF}_n(i)$ and $\mathrm{VIF}_n(i)$ is upper bounded by*

$$
\mathrm{Diff}(i) := \mathrm{VIF}_n(i) - \mathrm{IF}_n(i) = O_p(\frac{1}{n}).
$$

*Proof.* The difference between $\mathrm{IF}_n(i)$ and $\mathrm{VIF}_n(i)$ is given by

$$
\begin{aligned}
& \mathrm{Diff}_n(i) = \mathrm{VIF}_n(i) - \mathrm{IF}_n(i) \\
& = \left[ \nabla_\theta^2 \mathcal{L}(\hat\theta)/n \right]^{-1} \exp(\hat\theta^\top X_i) \cdot \frac{1}{n} \left\{ \sum_{j=1}^n \frac{I(Y_j \leq Y_i)\Delta_j}{S_n^{(0)}(Y_j; \hat\theta)} \cdot \left( X_i - \frac{S_n^{(1)}(Y_j; \hat\theta)}{S_n^{(0)}(Y_j; \hat\theta)} \right) \right. \\
& \qquad \left. - \sum_{j=1}^n \frac{I(Y_j < Y_i)\Delta_j}{S_n^{(0)}(Y_j; \hat\theta) - \exp(\hat\theta^\top X_i)/n} \cdot \left( X_i - \frac{S_n^{(1)}(Y_j; \hat\theta)}{S_n^{(0)}(Y_j; \hat\theta)} \right) \right\} \\
& = \left[ \nabla_\theta^2 \mathcal{L}(\hat\theta)/n \right]^{-1} \exp(\hat\theta^\top X_i) \cdot \frac{1}{n} \left\{ \sum_{j=1}^n \frac{I(Y_j \leq Y_i)\Delta_j}{S_n^{(0)}(Y_j; \hat\theta)} \cdot \left( X_i - \frac{S_n^{(1)}(Y_j; \hat\theta)}{S_n^{(0)}(Y_j; \hat\theta)} \right) \right. \\
& \qquad \left. - \sum_{j=1}^n \frac{I(Y_j \leq Y_i)\Delta_j}{S_n^{(0)}(Y_j; \hat\theta) - \exp(\hat\theta^\top X_i)/n} \cdot \left( X_i - \frac{S_n^{(1)}(Y_j; \hat\theta)}{S_n^{(0)}(Y_j; \hat\theta)} \right) \right\} \\
& \qquad + \frac{\Delta_i}{S_n^{(0)}(Y_i; \hat\theta) - \exp(\hat\theta^\top X_i)/n} \cdot \left( X_i - \frac{S_n^{(1)}(Y_i; \hat\theta)}{S_n^{(0)}(Y_i; \hat\theta)} \right) \Bigg) \\
& = - \left[ \nabla_\theta^2 \mathcal{L}(\hat\theta)/n \right]^{-1} \frac{\exp(2\hat\theta^\top X_i)}{n} \cdot \frac{1}{n} \sum_{j=1}^n \left\{ \frac{I(Y_j \leq Y_i)\Delta_j}{S_n^{(0)}(Y_j; \hat\theta)} \cdot \frac{1}{S_n^{(0)}(Y_j; \hat\theta) - \exp(\hat\theta^\top X_i)/n} \cdot \left( X_i - \frac{S_n^{(1)}(Y_j; \hat\theta)}{S_n^{(0)}(Y_j; \hat\theta)} \right) \right\} \\
& \qquad + \left[ \nabla_\theta^2 \mathcal{L}(\hat\theta)/n \right]^{-1} \frac{\exp(\hat\theta^\top X_i)}{n} \cdot \frac{\Delta_i}{S_n^{(0)}(Y_i; \hat\theta) - \exp(\hat\theta^\top X_i)/n} \cdot \left( X_i - \frac{S_n^{(1)}(Y_i; \hat\theta)}{S_n^{(0)}(Y_i; \hat\theta)} \right).
\end{aligned}
$$

Define

$$
J_n(t; \theta, Z_i) = \frac{I(t \leq Y_i)}{S_n^{(0)}(t; \theta)} \cdot \frac{1}{S_n^{(0)}(t; \theta) - \exp(\theta^\top X_i)/n} \cdot \left( X_i - \frac{S_n^{(1)}(t; \theta)}{S_n^{(0)}(t; \theta)} \right),
$$

and

$$
J(t; \theta, Z_i) = \frac{I(t \leq Y_i)}{s^{(0)}(t; \theta)} \cdot \frac{1}{s^{(0)}(t; \theta) - \exp(\theta^\top X_i)/n} \cdot \left( X_i - \frac{s^{(1)}(t; \theta)}{s^{(0)}(t; \theta)} \right).
$$

Then we rewrite $\mathrm{Diff}_n(i)$ using the empirical process notation:

$$
\mathrm{Diff}_n(i) = - \left[\nabla_\theta^2 \mathcal{L}(\hat{\theta})/n\right]^{-1} \frac{\exp(2\hat{\theta}^\top X_i)}{n} \cdot \int_0^\tau J_n(t; \hat{\theta}, Z_i) dG_n(t)
$$

$$
+ \left[\nabla_\theta^2 \mathcal{L}(\hat{\theta})/n\right]^{-1} \frac{\exp(\hat{\theta}^\top X_i)}{n} \cdot \frac{\Delta_i}{S_n^{(0)}(Y_i; \hat{\theta}) - \exp(\hat{\theta}^\top X_i)/n} \cdot \left(X_i - \frac{S_n^{(1)}(Y_i; \hat{\theta})}{S_n^{(0)}(Y_i; \hat{\theta})}\right). \tag{16}
$$

Next, we show that

$$
\int_0^\tau J_n(t; \hat{\theta}, Z_i) dG_n(t) = \int_0^\tau J(t; \theta^*, Z_i) dG(t) + o_p(1). \tag{17}
$$

To prove Eq. (17), we further decompose it into four terms:

$$
\int_0^\tau J_n(t; \hat{\theta}, Z_i) dG_n(t) - \int_0^\tau J(t; \theta^*, Z_i) dG(t) = \underbrace{\int_0^\tau \left(J_n(t; \hat{\theta}, Z_i) - J(t; \hat{\theta}, Z_i)\right) d(G_n(t) - G(t))}_{I_1}
$$

$$
+ \underbrace{\int_0^\tau J(t; \hat{\theta}, Z_i) d(G_n(t) - G(t))}_{I_2} + \underbrace{\int_0^\tau \left(J_n(t; \hat{\theta}, Z_i) - J(t; \hat{\theta}, Z_i)\right) dG(t)}_{I_3}
$$

$$
+ \underbrace{\int_0^\tau \left(J(t; \hat{\theta}, Z_i) - J(t; \theta^*, Z_i)\right) dG(t)}_{I_4}.
$$

For the first term $I_1$, by the triangle inequality, we have

$$
\sup_{t \in [0,\tau], \theta \in \mathcal{B}} \|J_n(t; \theta, Z_i) - J(t; \theta, Z_i))\|
$$

$$
\leq \sup_{t \in [0,\tau], \theta \in \mathcal{B}} \left\| \frac{I(t \leq Y_i)}{S_n^{(0)}(t; \theta)\left(S_n^{(0)}(t; \theta) - \exp(\theta^\top X_i)/n\right)} \cdot X_i - \frac{I(t \leq Y_i)}{\left[(s^{(0)}(t; \theta)\right]^2} \cdot X_i \right\|
$$

$$
+ \sup_{t \in [0,\tau], \theta \in \mathcal{B}} \left\| \frac{I(t \leq Y_i)}{\left[S_n^{(0)}(t; \theta)\right]^2 \left(S_n^{(0)}(t; \theta) - \exp(\theta^\top X_i)/n\right)} \cdot S_n^{(1)}(t; \theta) - \frac{I(t \leq Y_i)}{\left[(s^{(0)}(t; \theta)\right]^3} s^{(1)}(t; \theta) \right\|
$$

$$
\lesssim \sup_{t \in [0,\tau], \theta \in \mathcal{B}} \left| \frac{1}{\left[(S_n^{(0)}(t; \theta)\right]^2} - \frac{1}{\left[(s^{(0)}(t; \theta)\right]^2} \right| + O_p(\frac{1}{n})
$$

$$
+ \sup_{t \in [0,\tau], \theta \in \mathcal{B}} \left\| \frac{1}{\left[S_n^{(0)}(t; \theta)\right]^3} \cdot S_n^{(1)}(t; \theta) - \frac{1}{\left[(s^{(0)}(t; \theta)\right]^3} s^{(1)}(t; \theta) \right\| \tag{18}
$$

where the second inequality relies on the the boundedness of the support of $X_i$, $\tau$, and $\mathcal{B}$. Here, "$W_1 \lesssim W_2$" denotes that there exists a universal constant $C > 0$ such that $W_1 \leq CW_2$. Under Conditions (1)-(3), the function class $\{f_{t,\theta}(x, y) = I(y \geq t)\exp(\theta^\top x) : t \in [0, \tau], \theta \in \mathcal{B}\}$ is a Glivenko-Cantelli class, i.e., $\sup_{t \in [0,\tau], \theta \in \mathcal{B}} \|S_n^{(0)}(t; \theta) - s^{(0)}(t; \theta)\| = o_p(1)$. Similarly, we have $\sup_{t \in [0,\tau], \theta \in \mathcal{B}} \|S_n^{(1)}(t; \theta) - s^{(1)}(t; \theta)\| = o_p(1)$. By applying Taylor expansion to terms in Eq. (18) and the boundedness, we obtain the uniform convergence:

$$
\sup_{t \in [0,\tau], \theta \in \mathcal{B}} \|J_n(t; \theta, Z_i) - J(t; \theta, Z_i))\| = o_p(1). \tag{19}
$$

By the empirical process theory, we have $\sqrt{n}(G_n(t) - G(t))$ converges to a Gaussian process uniformly. Therefore, it follows that

$$
I_1 = \int_0^\tau \left(J_n(t; \hat{\theta}, Z_i) - J(t; \hat{\theta}, Z_i)\right) d(G_n(t) - G(t)) = o_p(1/\sqrt{n}).
$$

For the second term, note that $J(t; \hat{\theta}, Z_i)$ is bounded and thereby $I_2 = O_p(1/\sqrt{n})$. For the third term, due to uniform convergence in Eq. (19), it follows that $I_3 = o_p(1)$. Given the boundedness and the consistency of $\hat{\theta}$, i.e., $\hat{\theta} = \theta^* + o_p(1)$, we have $\sup_{t \in [0, \tau]} \|J(t; \hat{\theta}, Z_i) - J(t; \theta^*, Z_i)\| = o_p(1)$ and thereby $I_4 = o_p(1)$. So far, we have completed the proof of Eq. (17).

Finally, we plug in Eq. (17) together with known consistency results into Eq. (16): $\hat{\theta} = \theta^* + o_p(1)$ and $\nabla_\theta^2 \mathcal{L}(\hat{\theta})/n = \boldsymbol{A} + o_p(1)$, and obtain that

$$
\begin{aligned}
\mathrm{Diff}(i) = & -[\boldsymbol{A} + o_p(1)]^{-1} \frac{\exp(2\theta^{*\top} X_i) + o_p(1)}{\boldsymbol{n}} \cdot \left( \int_0^\tau J(t; \theta^*, Z_i) dG(t) + o_p(1) \right) \\
& + [\boldsymbol{A} + o_p(1)]^{-1} \frac{\exp(\theta^{*\top} X_i) + o_p(1)}{\boldsymbol{n}} \cdot \frac{\Delta_i}{s^{(0)}(Y_i; \theta^*) + o_p(1)} \left( X_i - \frac{s^{(1)}(Y_i; \theta^*) + o_p(1)}{s^{(0)}(Y_i; \theta^*) + o_p(1)} \right) \\
= & -[\boldsymbol{A}]^{-1} \frac{\exp(2\theta^{*\top} X_i)}{\boldsymbol{n}} \cdot \int_0^\tau J(t; \theta^*, Z_i) dG(t) \\
& + [\boldsymbol{A}]^{-1} \frac{\exp(\theta^{*\top} X_i)}{\boldsymbol{n}} \cdot \frac{\Delta_i}{s^{(0)}(Y_i; \theta^*)} \left( X_i - \frac{s^{(1)}(Y_i; \theta^*)}{s^{(0)}(Y_i; \theta^*)} \right) + o_p(\frac{1}{n}) \\
= & O_p(\frac{1}{n}).
\end{aligned}
$$

The second equality holds by the continuous mapping theorem and the third equality holds due to the boundedness of the support of $X$, $\mathcal{B}$, and $\tau$. We used the fact that there exists a positive constant $C > 0$ such that $\inf_{t \in [0, \tau], \theta \in \mathcal{B}} s^{(0)}(t; \theta) = \mathbb{E}\left( I(Y \geq t) \exp\left( \theta^\top X \right) \right) \geq C$. This completes the proof. $\square$

## B  DETAILED EXPERIMENT SETUP

**Datasets.** For Cox regression, both METABRIC and SUPPORT datasets are split into training, validation, and test sets with a 6:2:2 ratio. The training objects and test objects are defined as the full training and test sets. For node embedding, the test objects are all valid pairs of nodes, i.e., $34 \times 34 = 1156$ objects, while the training objects are the 34 individual nodes. In the case of listwise learning-to-rank, we sample 500 test samples from the pre-defined test set as the test objects. For the Mediamill dataset, we use the full label set as the training objects, while for the Delicious dataset, we sample 100 labels from the full label set (which contains 983 labels in total). The brute-force leave-one-out retraining follows the same training hyperparameters as the full model, with one training object removed at a time.

| Scenario | Dataset | Training obj | Test obj |
|---|---|---|---|
| Cox regression | METABRIC | 1217 samples | 381 samples |
| | SUPPORT | 5677 samples | 1775 samples |
| Node embedding | Karate | 34 nodes | 1156 pairs of nodes |
| Listwise learning-to-rank | Mediamill | 101 labels | 500 samples |
| | Delicious | 100 labels | 500 samples |

Table 4: Training objects and test objects in different experiment settings.

**Models.** For Cox regression, we train a CoxPH model with a linear function on the features for both the METABRIC and SUPPORT datasets. The model is optimized using the Adam optimizer with a learning rate of 0.01. We train the model for 200 epochs on the METABRIC dataset and 100 epochs on the SUPPORT dataset. For node embedding, we sample 1,000 walks per node, each with a length of 6, and set the window size to 3. The dimension of the node embedding is set to 2. For listwise learning-to-rank, the model is optimized using the Adam optimizer with a learning rate of 0.001, weight decay of 5e-4, and a batch size of 128 for 100 epochs on both the Mediamill and Delicious datasets. We also use TruncatedSVD to reduce the feature dimension to 8.

## C EFFICIENT INVERSE HESSIAN APPROXIMATION

Existing methods for efficient inverse Hessian approximation used by the conventional IF for decomposable losses can be adapted to accelerate VIF. Specifically, we consider two methods used by Koh & Liang (2017), Conjugate Gradient (CG) and LiSSA (Agarwal et al., 2017). The application of CG to VIF is straightforward, as it can be directly applied to the original Hessian matrix. LiSSA is originally designed for decomposable losses in the form $\sum_{i=1}^{n} \ell_i(\theta)$ and it accelerates the inverse Hessian vector product calculation by sampling the Hessians of individual loss terms, $\nabla_\theta^2 \ell_i(\theta)$. The adaptation of LiSSA to VIF depends on the specific form of the loss function. In many non-decomposable losses (e.g., the all three examples in this paper), the total loss can still be written as the summation of simpler loss terms, even though they are not decomposable to individual data points. In such cases, LiSSA can still be applied to efficiently estimate the inverse Hessian vector product through sampling the simpler loss terms.

### C.1 EXPERIMENTS

We implement the CG and LiSSA versions of accelerated VIF for the Cox regression model, and experiment them on the METABRIC dataset. In addition to the linear model, we also experiment with a neural network model, where the relative risk function is implemented as a two-layer MLP with ReLU activation. We use *VIF (Explicit)* to refer to the VIF with explicit inverse Hessian calculation, while using *VIF (CG)* and *VIF (LiSSA)* to refer to the accelerated variants.

**Performance.** As can be seen from Table 5, the accelerated methods VIF (CG) and VIF (LiSSA) achieve similar performance as both the original VIF (Explicit) and the Brute-Force LOO on both the linear and neural network models. The correlation coefficients of all methods on the neural network model are lower than those on the linear model due to the randomness inherent in the model training.

Table 5: The Pearson correlation coefficients of methods for Cox regression on the METABRIC dataset.

| Methods \ Models | Linear | Neural Network |
|---|---|---|
| VIF (Explicit) | 0.997 | 0.238 |
| VIF (CG) | 0.997 | 0.201 |
| VIF (LiSSA) | 0.981 | 0.197 |
| Brute-Force | 0.997 | 0.219 |

**Runtime.** We further report the runtime of different methods on neural network models with varying model sizes. VIF (CG) and VIF (LiSSA) are not only faster than VIF (Explicit), especially as the model size grows, but also much more memory efficient. VIF (Explicit) runs out of memory quickly as the model size grows, while VIF (CG) and VIF (LiSSA) can be scaled to much larger models.

Table 6: Runtime comparison of methods for Cox regression on the METABRIC dataset. The "#Param" refers to the total number of parameters in the neural network model.

| #Param | VIF (Explicit) | VIF (CG) | VIF (LiSSA) | Brute-Force |
|---|---|---|---|---|
| 0.04K | 9.88s | 5.68s | 8.85s | 5116s |
| 10.3K | 116s | 27.7s | 17.18s | 6289s |
| 41.0K | OOM | 113s | 67.7s | / |
| 81.9K | OOM | 171s | 79.1s | / |

## D HEATMAP OF NODE EMBEDDING

In Figure 2, we present the heatmap of the influence estimated by VIF and the actual LOO loss difference on two pairs of nodes. VIF could identify the top and bottom influential nodes accurately,

while the estimation of node influence in the middle more noisy. One caveat of these heatmap plots is that there is a misalignment between the color maps for VIF and LOO. This reflects the fact that, while VIF is effective at having a decent correlation with LOO, the absolute values tend to be misaligned.

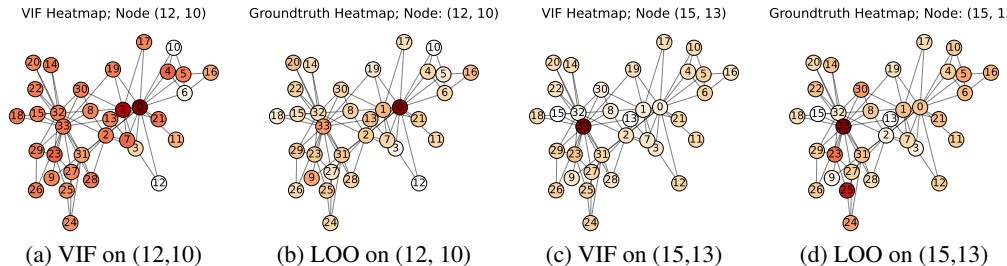

(a) VIF on (12,10)    (b) LOO on (12, 10)    (c) VIF on (15,13)    (d) LOO on (15,13)

Figure 2: VIF is applied to Zachary's Karate network to estimate the influence of each node on the contrastive loss of a pair of test nodes. Figure 2a and Figure 2b represent the heatmap of influence on the node pair (12,10). Figure 2c and Figure 2d represent the heatmap of influence on the node pair (15,13).

