# OpenReview forum: "A Versatile Influence Function for Data Attribution with Non-Decomposable Loss"
_ICLR.cc/2025/Conference — Submitted to ICLR 2025_

### Official Review · Reviewer_2PYn · 2024-10-22

**Soundness:** 2
**Presentation:** 3
**Contribution:** 2
**Rating:** 5
**Confidence:** 3

**Summary:**

The paper tries to generalize the influence function for cases where the used loss functions are non-decomposable, i.e., loss functions that can not be expressed as a sum of individual losses. To avoid deriving analytical formulas for calculating the limit in Eq. (7), the paper uses finite difference approximation method, which leads to their proposed versatile influence function (VIF).

**Strengths:**

The writing is clear, though there are some ambiguities in notation. I think tackling non-decomposable loss functions is a non-trivial generalization of the influence function.

**Weaknesses:**

Minor:
- In Lemma 1, strictly speaking,  Eq. (7) equivalently reduces to Eq. (3) as $\text{Eq. (7)} = n \times \text{Eq. (3)}$.

- The use of $\mathcal{L}(\theta, P) = \mathcal{L}(\theta, \mathbf{1})$, i.e., Eq. (8), is somewhat confusing. For example, according to Eq. (2), $\mathcal{L}(\theta, \mathbf{1}) = \sum_{i=1}^{n}\ell_i(\theta)$, but $\mathcal{L}(\theta, P) = \frac{1}{n} \sum_{i=1}^{n}\ell_i(\theta)$ in lines 664-666. Besides, while $\mathcal{L}(\theta, P)$ is understandable, the used instances, e.g., Eqs. (4) and (6), do not contain a data distribution $P$, and $\mathcal{L}(\theta, \mathbf{1})$ is more suitable. Therefore, I would suggest treating $P$ as an (unnormalized) weight vector instead of a data distribution for consistency in notation, and there might be better solutions.

Major:
- Lemma 2 lacks a proof, and I did not see why it holds. Substituting the defined $Q$ in what is in line 683 yields $\frac{1}{n-1}\sum_{1\leq j\leq n, j\not= i} \nabla_{\theta}\mathcal{L}(\hat{\theta}, z_j)$, whereas the result is $\nabla_{\theta}\mathcal{L}(\hat{\theta}, z_i)$ if we use $\delta_{z_i}$ instead.  This makes me feel $IF(\hat{\theta}(P); Q) \not= -IF(\hat{\theta}(P); \delta_{z_i})$, meaning that Lemma 2 is invalid.

- I feel the assumptions of strictly convexity and obtainable optimal solutions are too strong. Note that Bae et al. (2022) has addressed these two issues with the influence function, but their focus is on decomposable loss functions. With respect to non-decomposable loss functions, it would be beneficial for this work if these strong assumptions could be mitigated in some way.

- For the practical aspect, a potential drawback of the influence function (IF) is its need to calculate the inverse of Hessian matrices, which could be computationally expensive. It is clear that the proposed VIF also suffers from this issue which may limit its application to large scale settings. For decomposable loss functions, the corresponding Hassian matrices can be approximated by a sum of outer products, and then the calculation of inverse can be approximately simplified; see (Kwon et al., 2024). However, from my perspective, it is very likely that this technique is not applicable to non-decomposable losses.

Overall, I assess this work in two aspects. First, for the derivation of the proposed VIF, it is a straightforward adaptation of IF that makes me feel the contribution in methodology is minimal — nothing surprising. In particular, the imposed strong assumptions obscure why VIF is good. Second, I did not find any noticeable contributions to its practical deployment; the paper does not dicuss what if $\theta$ is large, and the conducted experiments only invlove small-scale settings as mentioned in lines 398-406.

Bae, J., Ng, N., Lo, A., Ghassemi, M., & Grosse, R. B. (2022). If influence functions are the answer, then what is the question?. Advances in Neural Information Processing Systems, 35, 17953-17967.

Kwon, Y., Wu, E., Wu, K., & Zou, J. (2024). DataInf: Efficiently Estimating Data Influence in LoRA-tuned LLMs and Diffusion Models. In The Twelfth International Conference on Learning Representations.

**Questions:**

Please refer to my comments on the weaknesses.

---

> ### Author Response · Authors · 2024-11-27
>
> We thank the reviewer for the insightful comments and suggestions. We address your concerns point-to-point below.
>
> > Minor points and Lemma 2
>
> We appreciate the careful and constructive feedback by the reviewer. We acknowledge that we missed some constants in Lemma 1 and Lemma 2 in the original submission, and the notations might be misleading due to heavy overloading.
>
> To address these concerns, We have **largely rewritten Section 3.4**. We have also systematically revised our notations throughout the paper to avoid notation overloading.
>
> **Lemma 1**: We have corrected the missing constant n. This lemma is now labeled as **Lemma 3.1**.
>
> **Lemma 2**: We have included Lemma 2 as part of the **new Theorem 3.1**. The proof of which is provided in Appendix A.3. A key step in this proof is to utilize the fact that $\sum_{j=1}^n \nabla_{\theta} l(\hat{\theta}, z_j) = 0$, which implies that $\sum_{j=1, j\neq i}^n \nabla_{\theta} l(\hat{\theta}, z_j) = -\nabla_{\theta} l(\hat{\theta}, z_i)$.
>
> **The connection between $\mathcal{L}(\theta, P)$ and $\mathcal{L}(\theta, 1)$**: We have provided a formal construction to connect these two loss definitions in the **new Proposition 3.1**. Specifically, for an empirical distribution on a (subset) of the n data points, we map the distribution to its support set, which corresponds to a binary vector based on whether each data point is present in this support set. The connection between the distribution and the binary vector is done through this mapping.
>
>
> > Efficient approximation of Hessian inverse
>
> We agree that mitigating the computational cost of Hessian inverse is critical for practical applications. To address this concern, we have explored the application of efficient inverse Hessian approximation methods to VIF, including Conjugate Gradient (CG) and LiSSA employed by Koh and Liang (2017) [1]. We also experimented them on the Cox regression model.
>
> The table below shows the Pearson correlation coefficients of different methods for Cox regression on the METABRIC dataset. We experimented with both linear and neural-network-based Cox regression. The efficient approximation methods, VIF (CG) and VIF (LiSSA) achieve similar performance as both the original VIF (Explicit) that explicitly inverts the Hessian and the Brute-Force LOO. The correlation coefficients of all methods on the neural network model are lower than those on the linear model due to the randomness inherent in the model training.
>
> | Methods / Models | Linear | Neural Network |
> |------------------|--------|--------|
> | VIF (Explicit)   | 0.997  | 0.238  |
> | VIF (CG)     	| 0.997  | 0.201  |
> | VIF (LiSSA)  	| 0.981  | 0.197  |
> | Brute-Force  	| 0.997  | 0.219  |
>
> We further report the runtime of different methods on neural network models with varying model sizes. VIF (CG) and VIF (LiSSA) are not only faster than VIF (Explicit), especially as the model size grows, but also much more memory efficient. VIF (Explicit) runs out of memory quickly as the model size grows, while VIF (CG) and VIF (LiSSA) can be scaled to much larger models.
>
> | \#Param | VIF (Explicit) | VIF (CG) | VIF (LiSSA) | Brute-Force |
> |---------|----------------|----------|-------------|-------------|
> | 0.1K	| 9.54s      	| 7.19s	| 12.5s   	| 5120s   	|
> | 10.3K   | 116s       	| 27.7s	| 17.18s  	| 6289s   	|
> | 41.0K   | OOM        	| 113s 	| 67.7s   	| /      	|
> | 81.9K   | OOM        	| 171s 	| 79.1s   	| /     	|
>
> These results are also added to a new Appendix C.
>
>
> **Technical details of VIF (LiSSA)**: We note that LiSSA is originally designed for decomposable losses in the form $\sum_{i=1}^n \ell_i(\theta)$ and it accelerates the inverse Hessian vector product calculation by sampling the Hessians of individual loss terms, $\nabla_{\theta}^2\ell_i(\theta)$. The adaptation of LiSSA to VIF depends on the specific form of the loss function. In many non-decomposable losses (e.g., the all three examples in this paper), the **total loss can still be written as the summation of simpler loss terms**, even though they are not decomposable to individual data points. In such cases, LiSSA can still be applied to efficiently estimate the inverse Hessian vector product by sampling the simpler loss terms.
>
> **Applicability of other techniques such as Kwon et al. (2024)**: For non-decomposable losses that can be written as **a summation of simpler (although not decomposable to data point level) loss terms**, the Hessian matrix may be similarly approximated by the sum of outer products of these simpler loss terms, enabling tricks such as the one utilized by Kwon et al. (2024) to efficiently approximate the Hessian inverse.
>
> [1] Koh, P. W., & Liang, P. (2017, July). Understanding black-box predictions via influence functions. In International conference on machine learning (pp. 1885-1894). PMLR.

---

> > ### Author Response · Authors · 2024-11-27
> >
> > > Mitigating convexity and obtainable optimal solution
> >
> > In our response to the Hessian inverse approximation concern, we have seen that VIF can *empirically* perform well on neural network models, where the convexity and obtainable optimal solution assumptions do not hold. While we acknowledge this is an important issue to address, we leave more theoretical treatment of this issue to future work.
> >
> > > Technical contribution and practical development
> >
> > We appreciate the reviewer’s summary of their assessment.
> >
> > **Technical contribution**: We have made substantial revisions in our method section to better clarify the motivation and theoretical results of the proposed method. Notably, we have introduced **a formal characterization of the approximation error of VIF under the Cox regression model** in **Theorem 3.2**, where we showed that the approximation error is in the order of $O(1/n)$, which vanishes for large training data size $n$.
> >
> > **Practical development**: To further enhance the practicality of the proposed method, we have integrated CG and LiSSA to accelerate VIF. These accelerated VIF algorithms have been empirically validated for their effectiveness in neural network models, showcasing the applicability of our method to larger-scale and complex machine learning settings.
> >
> > Overall, we would like to highlight that this is the first study that extends IF-based data attribution methods to non-decomposable losses, and we have verified the effectiveness of the proposed method across 3 distinct machine learning tasks. Our updated draft further provides novel theoretical guarantees of the approximation error on the Cox regression model, as well as practical algorithms for scalability. We believe the proposed method could enable efficient data attribution for more diverse machine learning tasks.

---

> > > ### Author Response · Authors · 2024-11-30
> > > **Discussion Appreciated**
> > >
> > > Dear Reviewer **2PYn**,
> > >
> > > Thank you once again for your valuable feedback. As the discussion period draws to a close, we would like to follow up to confirm whether our response has effectively addressed your concerns.
> > >
> > > Specifically:
> > >
> > > 1) Thank you pointing out the issues in our notation and lemmas. There were indeed some constants missed in the statements and we have corrected them with formal proofs.
> > >
> > > 2) We have added extensions of our methods and conducted additional experiments to show that the convexity assumption can be relaxed in practice and the inverse Hessian in the proposed method can be efficiently approximated. Therefore the proposed methods can be made scalable for neural network models.
> > >
> > > If there is any further question, please feel free to let us know. Thank you for your time and consideration.
> > >
> > > Best regards,
> > >
> > > Authors of submission 7674: A Versatile Influence Function for Data Attribution with Non-Decomposable Loss

---

> ### Comment · Reviewer_2PYn · 2024-12-02
>
> Thank you for your response. My concerns have been addressed, so I will raise my score to 5. However, I still have some reservations regarding the contributions. While I acknowledge that the novelty of this work lies in adapting the influence function for non-decomposable losses, I feel that the contributions are somewhat limited, as I do not find anything particularly insightful for further studies. Precisely, in my view, Theorem 3.1, which focuses on decomposable losses, seems more like a paraphrase of the influence function. For decomposable losses, their linearity in $\epsilon$ makes it unsurprising that the influence function is equivalent to using finite differences.
>
> I also noticed that the authors have added Theorem 3.2 to bound the proposed versatile influence function and influence function for cox regression, but I chose not to check the proof of it as I, personally, do not think it is a particularly interesting result for future work. So, I will drop my confidence to 3.

---

### Official Review · Reviewer_NvHf · 2024-11-02

**Soundness:** 3
**Presentation:** 3
**Contribution:** 2
**Rating:** 6
**Confidence:** 3

**Summary:**

This paper provides one efficient influence function estimation procedure for non-decomposable loss beyond the M-estimator. The procedure is directly estimated from the definition of influence function under the first-order optimal condition and illustrates superior computational benefits against the brute-force while enjoying similar results.

**Strengths:**

-	The formulas of VIF proposed in such paper is general for handling non-decomposable loss. To the best of my knowledge, I do not see much discussion for computing data influence under such non-decomposable loss.

**Weaknesses:**

-	**Lack of empirical baselines**: Numerous data attribution methods exist beyond the brute-force LOO. Many of these are likely faster than the naïve brute-force and can provide insights into empirical performance are in attributing data importance, e.g., some retrained-based methods such as datamodels [Illyas et al. 2022] and Shapley-value based approaches are relevant alternatives. A more thorough empirical comparison would clarify how the proposed VIF method stands in relation to these approaches.
-	**Insufficient theoretical discussions**: The paper would benefit from additional theoretical analysis regarding VIF convergence to true influence values, such as error bounds. Additionally, a comparison of VIF’s time complexity benefits—specifically, its matrix approximation efficiency relative to other methods and brute-force approaches—would strengthen the contribution. Given that VIF relies on approximations from influence functions, the authors should use terminology carefully in Sections 3.4 and 3.5, especially concerning terms like “empirically negligible” when attributing the target function and deriving formulas for Cox regression.
-	**Missing (explicit) conditions**: The paper would be improved by an explicit discussion of VIF’s limitations with respect to deep models or a clearer indication of when VIFs are applicable. The derivations depend on the first-order optimality condition (Line 263), inheriting limitations from prior work [Koh and Liang 2017]. Given that deep models often do not satisfy this condition [Basu et al. 2021], it is unclear how the influence-based calculations are expected to perform in such settings.


Reference:
[Basu et al. 2021] Influence Functions in Deep Learning Are Fragile. ICLR 2021.

**Questions:**

-	The authors need to be more rigorous for the difference in literature. I do not understand why the difference of denominator between IF (derived for Cox regression) and VIF in Line 380 – 386 can be ignored. A detailed comparison between these would be necessary Furthermore, a clear definitions of $\epsilon_{ij}$ means in this context would be helpful.
-	The authors can benefit from providing additional examples of non-decomposable losses to illustrate the advantages of expressing the influence function in the form of (10) – where the loss difference is first computed and then the gradient is taken, as opposed to directly using the inverse. Note that M-estimators should not be included here as one example, as they inherently have straightforward forms, as mentioned in Section 3.5.

---

> ### Author Response · Authors · 2024-11-27
>
> We thank the reviewer for the insightful comments and suggestions. We address your concerns point-to-point below.
>
> > Weakness 1: Empirical baselines such as retraining-based methods
>
> We acknowledge that retraining-based methods can indeed be applied for non-decomposable losses. However, such methods are usually orders of magnitude slower than influence function style methods. In fact, for Data Shapley, there is a known lower bound O(n log n) on the number of model retraining required for a given approximation error [1]. Therefore it will be even slower than brute-force LOO for large n.
>
> Moreover, in this study, we have used LOO as the ground truth for evaluation. Retraining-based methods such as Data Shapley are NOT designed to optimize for LOO, so they are UNLIKLEY to perform very well according to our evaluation metric, which is based on LOO correlation. While it is debatable whether LOO is always desirable as the data attribution target, arguably this is the most widely used criterion in the literature. Approximating data attribution targets other than LOO (such as Shapley value) is beyond the scope of this paper.
>
> As a result, we consider comparisons with retraining-based methods unnecessary. Nevertheless, we carried out one additional experiment with Cox regression on the METABRIC dataset using Data Banzhaf [2], a retraining-based method close to Data Shapley (we have initially tried Data Shapley but our implementation failed to converge and we are still investigating the reason). This method took 20 min (LOO: 24 min, VIF: 2.43 sec), while achieving an LOO correlation of 0.657 (LOO: 0.997, VIF: 0.997).
>
> Another empirical baseline we experimented with is a naive application of the conventional influence function on the Cox model. In this case, we pretend that the summation in Eq (4) is a decomposable loss, ignoring the impact of the interaction among data points in the loss. This naive baseline achieves an LOO correlation of 0.537 (LOO: 0.997, VIF: 0.997) on the METABRIC dataset. This result highlights the need for a principled design to approximate LOO on non-decomposable loss.
>
> [1] Wang, J. T., & Jia, R. (2023). A Note on" Towards Efficient Data Valuation Based on the Shapley Value''. arXiv preprint arXiv:2302.11431.
>
> [2] Wang, J. T., & Jia, R. (2023, April). Data banzhaf: A robust data valuation framework for machine learning. In International Conference on Artificial Intelligence and Statistics (pp. 6388-6421). PMLR.
>
> > Weakness 2 and Question 1: Theoretical discussions
>
> We appreciate the reviewer’s feedback. To address the reviewer’s concern, we have largely rewritten Section 3.4 and 3.5 to make our statements more rigorous. Specifically to one point mentioned by the reviewer, we have provided **a formal characterization of the approximation error of VIF under the Cox regression model** in **Theorem 3.2**, where we showed that the approximation error is in the order of $O(1/n)$, which vanishes for large training data size $n$.
>
> Regarding computational efficiency, we have explored the application of efficient inverse Hessian approximation methods to VIF, including Conjugate Gradient (CG) and LiSSA employed by Koh and Liang (2017) [1]. We also experimented them on the Cox regression model.
>
> The table below shows the Pearson correlation coefficients of different methods for Cox regression on the METABRIC dataset. We experimented with both linear and neural-network-based Cox regression. The efficient approximation methods, VIF (CG) and VIF (LiSSA) achieve similar performance as both the original VIF (Explicit) that explicitly inverts the Hessian and the Brute-Force LOO. The correlation coefficients of all methods on the neural network model are lower than those on the linear model due to the randomness inherent in the model training.
>
> | Methods / Models | Linear | Neural Network |
> |------------------|--------|--------|
> | VIF (Explicit)   | 0.997  | 0.238  |
> | VIF (CG)     	| 0.997  | 0.201  |
> | VIF (LiSSA)  	| 0.981  | 0.197  |
> | Brute-Force  	| 0.997  | 0.219  |
>
> We further report the runtime of different methods on neural network models with varying model sizes. VIF (CG) and VIF (LiSSA) are not only faster than VIF (Explicit), especially as the model size grows, but also much more memory efficient. VIF (Explicit) runs out of memory quickly as the model size grows, while VIF (CG) and VIF (LiSSA) can be scaled to much larger models.
>
> | \#Param | VIF (Explicit) | VIF (CG) | VIF (LiSSA) | Brute-Force |
> |---------|----------------|----------|-------------|-------------|
> | 0.1K	| 9.54s      	| 7.19s	| 12.5s   	| 5120s   	|
> | 10.3K   | 116s       	| 27.7s	| 17.18s  	| 6289s   	|
> | 41.0K   | OOM        	| 113s 	| 67.7s   	| /      	|
> | 81.9K   | OOM        	| 171s 	| 79.1s   	| /     	|
>
> These results are also added to a new Appendix C.
>
> [1] Koh, P. W., & Liang, P. (2017, July). Understanding black-box predictions via influence functions. In International conference on machine learning (pp. 1885-1894). PMLR

---

> > ### Author Response · Authors · 2024-11-27
> >
> > > Weakness 3: Limitations with respect to deep models
> >
> > We thank the reviewer for the suggestion. We have now **included a limitation discussion at the end of the paper**.
> >
> > The convexity assumption is the main limitation in the formal derivation of the VIF, similar to Koh and Liang (2017) [1]. However, as shown in our response to the previous point, we have successfully adapted CG and LiSSA to VIF. We believe that more advanced techniques to stabilize and accelerate IF-based methods developed for decomposable losses, such as EK-FAC [2], ensemble [3], or gradient projection [4], may be adapted to further enhance the practical applicability of VIF on large-scale models.
> >
> > [1] Koh, P. W., & Liang, P. (2017, July). Understanding black-box predictions via influence functions. In International conference on machine learning (pp. 1885-1894). PMLR.
> >
> > [2] Grosse, R., Bae, J., Anil, C., Elhage, N., Tamkin, A., Tajdini, A., ... & Bowman, S. R. (2023). Studying large language model generalization with influence functions. arXiv preprint arXiv:2308.03296.
> >
> > [3] Park, S. M., Georgiev, K., Ilyas, A., Leclerc, G., & Madry, A. (2023). Trak: Attributing model behavior at scale. arXiv preprint arXiv:2303.14186.
> >
> > [4] Choe, S. K., Ahn, H., Bae, J., Zhao, K., Kang, M., Chung, Y., ... & Xing, E. (2024). What is Your Data Worth to GPT? LLM-Scale Data Valuation with Influence Functions. arXiv preprint arXiv:2405.13954.
> >
> > > Question 2: The benefit of calculating the loss difference first
> >
> > The benefit essentially comes from the fact that, when calculating the VIF for the i-th data point, loss terms that do not involve this data point will cancel out. For example, the Cox regression model is also an example. While each loss term of the Cox regression model involves multiple data points, a single data point is typically only involved in a subset of the loss terms. Those loss terms that do not involve the data point for VIF calculation will be canceled out if we take the difference first.
> >
> > We have clarified this point in our comments about the Computational Advantages.

---

> > > ### Comment · Reviewer_NvHf · 2024-11-27
> > >
> > > Thank the author for the detailed feedback and revision to the original paper. It looks better to me overall! I still find some potentially further improvements during the revision:
> > > - $Q_{n - 1}$ had better need a subscript $i$.
> > > - Proposition 3.1 can be more clear. That is, I think it may be unnecessary to define $\tilde L$ and $L$ again. And the authors may want to highlight the definition of $\tilde L$ and $L$ formally in the structure of definitions.
> > > - Finite-difference $IF_{\varepsilon}$ (Def 3.2) can use a different notation, e.g. $\widehat{IF}$ or sth else to better differentiate with the original true one (equation 7).
> > > - It would be better to make more clear and highlight the different order of Reid & Crepeau's solution and the true influence against the current VIF, e.g. if $IF_{n}(i) - IF(i)$ is of order $1/\sqrt{n}$ (larger than $O(1/n)$), then such VIF approximation $O(1/n)$ is valuable.

---

> > > > ### Author Response · Authors · 2024-11-27
> > > >
> > > > We thank the reviewer for the further suggestions.
> > > >
> > > > For $Q_{n-1}$, we have changed it to $Q^{(-i)}_{n-1}$ to highlight its dependence on $i$. We have also added the hat for the finite-difference IF notation following the reviewer’s suggestion.
> > > >
> > > > Regarding Proposition 3.1 and the definitions of $\tilde{\mathcal{L}}$ and $\mathcal{L}$, we first clarify that $\mathcal{L}$ is defined in Definition 3.1. On the other hand, $\tilde{\mathcal{L}}$ can take different forms depending on the specific machine learning task and the choices of P and Q. We include a definition of $\tilde{\mathcal{L}}$ for general non-decomposable losses in Proposition 3.1. Note that this is to maintain consistency with Lemma 3.1, where the $\tilde{\mathcal{L}}$ is defined differently and specifically for M-estimation.
> > > >
> > > > Regarding Reid \& Crepeau, it is known that $IF_n(i)$ is a consistent estimator for $IF(i)$, i.e, $IF_n(i) = IF(i) + o_p(1)$ (see the Approximation Error paragraph in Appendix A.5), which indicates that $IF_n(i)$ is bounded away from $0$ in probability. Our Theorem 3.2 suggests that $VIF_n (i) = IF_n(i) +O_p(1/n)$, which has two implications. Firstly, $VIF_n (i) - IF_n(i)$ is in a smaller order than $IF_n(i)$, which implies that when using $VIF_n(i)$ as an approximation to $IF_n(i)$ for downstream data attribution, it has a negligible difference. Secondly, the proposed VIF is also a consistent estimator for $IF(i)$ under Cox regression. Furthermore, it is worth noting that the solution by Reid \& Crepeau is an analytical solution specifically tailored for Cox regression, while our VIF is a generic method that can be easily calculated through auto-differentiation.
> > > >
> > > > We agree that it would be even more valuable if we could further show the rate of convergence of $IF_n(i) - IF(i)$. Using similar techniques for our Theorem 3.2, we can show that $IF_n(i) - IF(i) = \Delta_i o_p(1) + o_p(1/\sqrt{n})$. But it does not imply a clear comparison between $VIF_n (i) - IF_n(i)$ and $IF_n(i) - IF(i)$. Therefore, we decide to leave this result out of our paper to avoid confusion for the readers.

---

> > > > > ### Comment · Reviewer_NvHf · 2024-11-28
> > > > >
> > > > > Thank you for the further clarification!
> > > > >
> > > > > For the definitions of $\tilde L$ and $L$, my suggestion is to put the definition in proposition 3.1 upfront in Sec 3.3 (since I thought it is a general definition aligned with that in Def 3.1) with the explanations in Line 255, where $M$-estimator in Lemma 3.1 is a special case. This could make the paper more coherent.
> > > > >
> > > > > Besides, I have no more major concerns about that paper.  Thanks for all the improvement and clarification!

---

> > > > > > ### Author Response · Authors · 2024-11-28
> > > > > >
> > > > > > Dear Reviewer,
> > > > > >
> > > > > > Thank you for the final suggestion. We will leverage it into our final version, as we can no longer edit our submission at this moment.
> > > > > >
> > > > > > We deeply appreciate the reviewer's super careful review of our paper and the subsequent discussions, which have substantially improved our paper compared to the original submission.
> > > > > >
> > > > > > We understand that the reviewer's initial rating was already positive--which we sincerely appreciate--and we do not anticipate a further change of the rating from the reviewer. However, if possible, we have one request for the reviewer: could you kindly **provide an objective summary of your evaluation for the AC to consider** during the decision-making process?
> > > > > >
> > > > > > We hope to have the opportunity to engage with other reviewers in the coming days. However, if further engagements do not take place, your opinion on our post-rebuttal draft will be especially critical.
> > > > > >
> > > > > > Thank you once again for your time and effort in reviewing our submission!

---

### Official Review · Reviewer_1htT · 2024-11-03

**Soundness:** 3
**Presentation:** 2
**Contribution:** 2
**Rating:** 6
**Confidence:** 3

**Summary:**

The authors propose a novel method for approximating the Leave One Out (LOO) data attribution scores. This is achieved by extending the influence-function-based attribution method into a more general class of loss functions. They highlight the fact that this formulation allows the computations to be approximated using finite differences, in turn allowing the influence function to be calculated using autodiff methods. Experiments are included to support the closeness of the approximations and to show the improvement in computational efficiency, when compared to the ordinary LOO computation.

**Strengths:**

1. The paper presents a novel formulation of a general class of losses (including non-decomposable losses, i.e., the losses which cannot be written as a sum of a function of each training example alone) which facilitates the application of an influence function for data attribution.
2. Provides analytical results justifying the accuracy of the finite difference approximation (VIF) of the influence function.
3. Experiments back-up the usefulness of VIF as a data attribution method, in terms of both accuracy and computational efficiency.
4. The presentation of the contents and the notation is generally well understandable.

**Weaknesses:**

1. I believe the selection of $\epsilon=1$ in the finite difference approximation of the IF is not sufficiently justified. Why not use any other $\epsilon$?
2. The VIF approximation is shown to be very close to the exact IF in two special cases. However the discussion on IF as an approximation for LOO (i.e., Eq. (7) being a good approximation of (3)) under the assumption of P and Q being uniform distributions over the corresponding samples with and without the object of interest is limited to M-estimators. Can anything be said on the Cox regression or any other model?
3. The paper will benefit from a limitations section which summarizes all the assumptions made along the way (e.g.: the convexity of the loss).

**Questions:**

Please see weaknesses.
A suggestion: Complementing Fig. 1, a heat-map of “influence estimated by VIF” and another heat-map of “actual difference in the contrastive loss” for a given pair of nodes, plotted on the graph would be more visually expressive.

---

> ### Author Response · Authors · 2024-11-27
>
> We thank the reviewer for the insightful comments and suggestions. We address your concerns point-to-point below.
>
> > Weakness 1: Justification of the finite difference
>
> We thank the reviewer for the valuable question. We have largely rewritten Section 3.4 to provide clearer motivation for the finite difference.
>
> In the updated draft, we have first formally defined the finite-difference influence function in **Definition 3.2**. Then we clearly stated the motivation for this finite-difference approximation in **Theorem 3.1**, where we demonstrated that, under the M-estimators, the statistical influence function involving a limit in Eq (7) **exactly equals to** a finite-difference influence function with a certain choice of $\varepsilon=-\frac{1}{n-1}$. We also showed that the finite difference approximation with $\varepsilon=1$ can be viewed as a “reparameterization” of the original $\varepsilon=-\frac{1}{n-1}$ finite-difference influence function. Specifically, the change from $\varepsilon=-\frac{1}{n-1}$ to $\varepsilon=1$ is a result of changing the perturbation direction $Q$ from $\delta_{z_i}$ to $\mathbb{Q}_{n-1}$. While $\varepsilon=1$ may appear large, $\mathbb{Q}_{n-1}$ is closer to $\mathbb{P}_n$ than $\delta_{z_i}, therefore the approximation is still good.
>
> In short, the selection of $\varepsilon=1$ is not an arbitrary choice–it comes from a careful choice of the perturbation direction $Q$, with a theoretical observation under the M-estimators. We refer the reviewer to **Theorem 3.1** for more details, which should better clarify the question of why this finite-difference approximation may approximate the original limit well.
>
> > Weakness 2: Approximation error on Cox regression
>
> In our updated draft, we have further provided **a formal characterization of the approximation error of VIF under the Cox regression model** in **Theorem 3.2**, where we showed that the approximation error is in the order of $O(1/n)$, which vanishes for large training data size $n$.
>
> > Weakness 3: Limitations
>
> We thank the reviewer for the suggestion. We have now **included a limitation discussion at the end of the paper**.
>
> The convexity assumption is the main limitation in the formal derivation of the VIF. However, it is possible to adapt computational tricks from existing IF-based methods for decomposable losses to VIF. For example, we have included a new **Appendix C** exploring the application of Conjugate Gradient and LiSSA employed by Koh and Liang (2017) [1] to VIF. We have also successfully implemented them on neural-network-based Cox regression models. Please see Appendix C for more details about this.
>
> [1] Koh, P. W., & Liang, P. (2017, July). Understanding black-box predictions via influence functions. In International conference on machine learning (pp. 1885-1894). PMLR.
>
> > Question 1: Heatmap
>
> We newly added **Appendix D** to visualize the heatmap of influence estimated by VIF and LOO for two pairs of nodes. VIF could identify the top and bottom influential nodes accurately, while the estimation of node influence in the middle more noisy. One caveat of these heatmap plots is that there is a misalignment between the color maps for VIF and LOO. This reflects the fact that, while VIF is effective at having a decent correlation with LOO, the absolute values tend to be misaligned.

---

> > ### Author Response · Authors · 2024-11-30
> > **Discussion Appreciated**
> >
> > Dear Reviewer **1htT**,
> >
> > We appreciate your positive initial rating and the valuable feedback. As the discussion period draws to a close, we would like to follow up to confirm whether our response has effectively addressed your concerns.
> >
> > Specifically, 1) we have significantly revised our Section 3.4 to provide better clarification for the finite difference choice; 2) we have provided a formal theoretical guarantee about the approximation error of VIF for Cox regression; 3) we have also added a limitation discussion in our draft.
> >
> > If there is any further question, please feel free to let us know. Thank you for your time and consideration.
> >
> > Best regards,
> >
> > Authors of submission 7674: A Versatile Influence Function for Data Attribution with Non-Decomposable Loss

---

> > > ### Comment · Reviewer_1htT · 2024-12-01
> > > **All concerns addressed well**
> > >
> > > Dear Authors,
> > >
> > > All the points I have raised have been addressed well by the authors. I appreciate the effort put into deriving/generating these new results. Thank you!

---

> > > > ### Author Response · Authors · 2024-12-02
> > > >
> > > > Thank you for acknowledging our response! We also appreciate your valuable feedback that greatly improved our paper.

---

### Official Review · Reviewer_UPcL · 2024-11-08

**Soundness:** 2
**Presentation:** 3
**Contribution:** 2
**Rating:** 5
**Confidence:** 3

**Summary:**

This work proposes an influence function (IF) for loss functions beyond M-estimators. The proposed IF is derived from an approximation of a general IF formulation. Its simplified form enables better computational efficiency, as verified through experiments.

**Strengths:**

The proposed idea is well-motivated and the presentation is good overall.

**Weaknesses:**

1. Some of the notations are incorrect. $P$ and $Q$ are probability measures defined on the sample space. $\boldsymbol{1}$ is a vector of ones, but not a probability measure. In (8), (9), and some other places, they look like letting $P$ be  $\boldsymbol{1}$.

2. The finite difference approximate is confusing. Why the limit of $\varepsilon \to 0$ can be approximated by $\varepsilon=1$? What does finite difference mean here as $\varepsilon \in [0,1]$? What are the theoretical guarantees about this approximation? This approximation is crucial for Definition 2 and the advantage of VIF. It should be properly justified.

**Questions:**

See weakness 2

---

> ### Author Response · Authors · 2024-11-27
>
> We thank the reviewer for the insightful comments and suggestions. We address your concerns point-to-point below.
>
> > Weakness 1: Notations of P, Q, and the all one vector.
>
> We clarify that we are not letting P be 1. There are two definitions of the loss function in our derivation, $\mathcal{L}(\theta, P)$ and $\mathcal{L}(\theta, b)$. The former depends on a probability measure, while the latter depends on a binary vector. In our original submission, we overloaded the notation $\mathcal{L}$ for the two definitions for convenience (we had a small footnote on page 3 in our original submission about this overloading). We also made some connection between the two definitions of the loss, and this may have led to the impression that we let P be 1 somewhere.
>
> After reviewing our writing, we acknowledge that overloading the notations in our original submission may indeed lead to confusion. To address the reviewer’s concern, we have thoroughly revised our notations in the updated draft, avoiding all notation overloading.
>
> In our updated draft, the loss function depending on the probability measure is now written as $\tilde{\mathcal{L}}$. Moreover, we have made it clearer and more rigorous about how we connect the two definitions of loss function in the new **Proposition 3.1**. Specifically, for an empirical distribution on a (subset) of the n data points, we map the distribution to its support set, which corresponds to a binary vector based on whether each data point is present in this support set. The connection between the distribution and the binary vector is done through this mapping.
>
> > Weakness 2: The motivation and theoretical guarantee of the finite difference
>
> We thank the reviewer for this valuable question. We have largely rewritten Section 3.4 to provide clearer motivation for the finite difference.
>
> In the updated draft, we have first formally defined the finite-difference influence function in **Definition 3.2**. Then we clearly stated the motivation for this finite-difference approximation in **Theorem 3.1**, where we demonstrated that, under the M-estimators, the statistical influence function involving a limit in Eq (7) **exactly equals to** a finite-difference influence function with a certain choice of $\varepsilon=-\frac{1}{n-1}$. We also showed that the finite difference approximation with $\varepsilon=1$ can be viewed as a “reparameterization” of the original $\varepsilon=-\frac{1}{n-1}$ finite-difference influence function. Specifically, the change from $\varepsilon=-\frac{1}{n-1}$ to $\varepsilon=1$ is a result of changing the perturbation direction $Q$ from $\delta_{z_i}$ to $\mathbb{Q}_{n-1}$. While $\varepsilon=1$ may appear large, $\mathbb{Q}_{n-1}$ is closer to $\mathbb{P}_n$ than $\delta_{z_i}, therefore the approximation is still good.
>
> We refer the reviewer to **Theorem 3.1** for more details, which should better clarify the question of why this finite-difference approximation may approximate the original limit well.
>
> Regarding theoretical guarantees, in our original submission, we have shown that the proposed VIF falls back to the original influence function under M-estimators (decomposable losses) without any approximation error. In our updated draft, we have further provided **a formal characterization of the approximation error of VIF under the Cox regression model** in **Theorem 3.2**, where we showed that the approximation error is in the order of $O(1/n)$, which vanishes for large training data size $n$.

---

> > ### Author Response · Authors · 2024-11-30
> > **Discussion Appreciated**
> >
> > Dear Reviewer UPcL,
> >
> > Thank you once again for your valuable feedback. As the discussion period draws to a close, we would like to follow up to confirm whether our response has effectively addressed your concerns.
> >
> > Specifically, 1) we believe your confusion of letting $P$ be $\mathbf{1}$ comes from the notation overloading of $\mathcal{L}$ in our original submission, and we have systematically revised our notations to avoid such overloading; 2) we have also provided detailed motivation for the finite difference in both our revised draft and our earlier response.
> >
> > If there is any further question, please feel free to let us know. Thank you for your time and consideration.
> >
> > Best regards,
> >
> > Authors of submission 7674: A Versatile Influence Function for Data Attribution with Non-Decomposable Loss

---

### Author Response · Authors · 2024-11-27
**Message to all reviewers**

We sincerely thank all the reviewers for their insightful comments and suggestions. We apologize for the delayed response as we have made substantial revisions to improve the clarity of our draft, added new theoretical guarantees for the proposed method, and developed efficient extensions of our method to enhance its scalability.

We have addressed all the comments in the detailed individual response to each reviewer. We have also updated the paper to reflect these changes.  Here we would like to highlight a few key points in our response.
- **Formal characterization of approximation error**
    - We have provided a formal characterization of the approximation error of VIF under the Cox regression model in the **new Theorem 3.2**.
    - The theorem showed that the approximation error is in the order of $O(1/n)$, which vanishes for large training data size $n$.
- **Enhanced clarity in Section 3.4**
    - **Section 3.4** has been substantially rewritten to better clarify the motivation behind the proposed method.
    - In particular, we have clarified **the choice of the finite difference** by formally characterizing the relationship between the proposed finite-difference approximation and the statistical IF in the **new Theorem 3.1**.
- **Efficient Hessian inverse approximation**
    - We have explored methods such as Conjugate Gradient and LiSSA for efficient Hessian inverse approximation for the proposed VIF
    - These approaches have shown strong performance on both linear and neural-network-based Cox regression models, with results provided in the new **Appendix C**.
- **Improved notations**
    - The notations throughout the paper have been systematically revised to eliminate overloading.

Once again, we thank the reviewers for their time and effort in evaluating our work. We are happy to provide additional information or address any further concerns.

---

### Meta-Review · Area_Chair_4GER · 2024-12-22

**Metareview:**

The paper presents a novel formulation for a general class of losses, including non-decomposable losses, which allows for the application of an influence function for data attribution. The paper provides analytical results to justify the accuracy of the finite difference approximation (VIF) of the influence function and includes experiments that support the usefulness of VIF as a data attribution method in terms of accuracy and computational efficiency

However, the reviewers also raised concerns about the novelty of the proposed VIF. They find Theorem 3.1 to be a "paraphrase" of existing influence function work and Theorem 3.2 "not particularly interesting." The paper initially relies on strong assumptions like strict convexity and easily obtainable optimal solutions, which might not hold in practice. While the authors attempt to address this by showing VIF works empirically on neural networks, they don't fully address this theoretically. Reviewer NvHf points out the lack of sufficient empirical comparisons with other data attribution methods beyond brute-force LOO.

For these reasons, overall, the reviewers felt the paper is slightly below the acceptance threshold in its current state.

**Additional Comments On Reviewer Discussion:**

- Authors rewrote Section 3.4 to better clarify the motivation for choosing ϵ=1 in the finite difference approximation.
- They provided a formal characterization of the approximation error of VIF under the Cox regression model in Theorem 3.2, showing the error to be in the order of O(1/n) .
- They included a limitations section in the paper, addressing concerns about the convexity assumption and discussing potential ways to mitigate this limitation.
- They added experiments with Data Banzhaf, a retraining-based method, and a naive application of the conventional influence function on the Cox model.
- They explored the application of efficient inverse Hessian approximation methods to VIF, including Conjugate Gradient (CG) and LISSA, and added experiments on the Cox regression model with both linear and neural-network-based architectures.

---

### Decision · Program_Chairs · 2025-01-22

Reject